# Score-Based Multimodal Autoencoders

## Abstract

Multimodal Variational Autoencoders (VAEs) represent a promising group of generative models that facilitate the construction of a tractable posterior within the latent space, given multiple modalities. Daunhawer et al. (2022) demonstrate that as the number of modalities increases, the generative quality of each modality declines. In this study, we explore an alternative approach to enhance the generative performance of multimodal VAEs by jointly modeling the latent space of unimodal VAEs using score-based models (SBMs). The role of the SBM is to enforce multimodal coherence by learning the correlation among the latent variables. Consequently, our model combines the superior generative quality of unimodal VAEs with coherent integration across different modalities.

## 1 Introduction

The real-world data often has multiple modalities such as image, text, and audio, which makes learning from multiple modalities an important task. Multimodal VAEs are a class of multimodal generative models that are able to generate multiple modalities jointly. Learning from multimodal data is inherently more challenging than from unimodal data, as it involves processing multiple data modalities with distinct characteristics.

In order to learn the joint representation of these modalities, previous approaches generally preferred to encode them to latent distribution that governs the data distribution across different modalities. In general, we expect the following properties from a multimodal generative model:

**Multiway conditional generation:** Given the presence of certain modalities, it should be feasible to generate the absent modalities based on the existing ones (Shi et al., 2019; Wu & Goodman, 2018). The conditioning process should not be limited to specific modalities; rather, any modality should serve as a basis for generating any other modality.

**Unconditional generation:** If we have no modality present to condition on, we should be able to sample from the joint distribution so that the generated modalities are coherent (Shi et al., 2019). Coherence in this case means that the generated modalities represent the same concept that is expressed in the different modalities we have.

**Conditional modality gain:** When additional information is provided to the model through the observation of more modalities, there should be an enhancement in the performance of the generated absent modalities. In other words, the generation performance should consistently improve as the number of observed (given) modalities increases.

**Scalability:** The model should scale as the number of total modalities increases. Moreover, the model complexity shouldn't become computationally inefficient when adding more modalities (Sutter et al., 2020).

Other properties like joint representation where we want a representation that takes into account the statistics and properties of all the modalities (Srivastava & Salakhutdinov, 2014; Suzuki et al., 2017) can be easily learned from a multimodal model obtaining the above properties. Another property that we have deferred for future work is weak supervision where we make use of multimodal data that aren't paired together (Wu & Goodman, 2018).

Naively using multimodal VAEs by training all combinations of modalities becomes easily intractable as the number of models to be trained increases exponentially. We will need to train $2^M - 1$ different combinations of models for each subset of the modalities. Since this is not a scalable approach,

previous works have proposed different methods of avoiding this by constructing a joint posterior of all the modalities. They do this by modeling the joint posterior over the latent space $\mathbf{z}$: $q(\mathbf{z}|\mathbf{x}_{1:M})$, where $\mathbf{x}_{1:M}$ is the set of modalities. To ensure the tractability of the inference network $q$, prior works have proposed using a product of experts (Wu & Goodman, 2018), mixture of experts (Shi et al., 2019), or in the generalized form, mixture of the product of experts (MoPoE) (Sutter et al., 2021); among the others.

After selecting on how to fuse the posteriors of different modalities, these approaches then construct the joint multimodal ELBO and use it to train multimodal data. At inference time, they use the modalities that are observed to generate the missing modalities. Wu & Goodman (2018) intentionally adds ELBO subsampling during training to increase the model's performance on generating missing modalities at inference time. Mixture of experts and mixture of products of experts subsample modalities as part of their training process because of the posterior model is a mixture model. Subsampling of the modalities, as pointed out by Daunhawer et al. (2022), results in a generative discrepancy among modalities. We also observe that conditioning on more modalities often reduces the quality of the generated modality which goes contrary to one's expectation. As a model receives additional information, it should perform better and better but this doesn't happen in previous multimodal VAEs as the generated modalities continue to have lower qualities as more modalities are observed. Daunhawer et al. (2022) further concludes that these models cannot be useful for real-world applications at this point due to these failures.

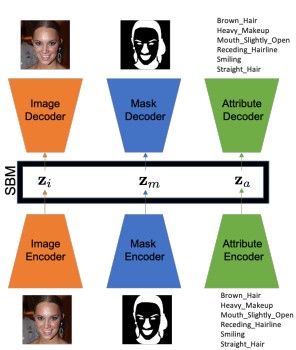

Figure 1: A variational or regularized auto-encoder will be used for each individual modality to get the latent representation which then will be used to train the score-based model which will allow the prediction of any modality given some or none. The auto-encoders are trained independently in the first stage and the respective $z$ of each modality will be used to train the score network.

To overcome these issues, instead of constructing a joint posterior, we try to explicitly model the joint latent space of individual VAEs: $p_\theta(\mathbf{z}_{1:M})$. The joint latent model learns the correlation among the individual latent space in a separate score-based model trained independently without constructing a joint posterior as the previous multimodal VAEs. Therefore, it can ensure prediction coherence by sampling from the score model while also maintaining the generative quality that is close to a unimodal VAE. And by doing so, we avoid the need to construct the joint multimodal ELBO. We use the independently trained unimodal VAEs to generate latent samples, and use those samples to train a score network that models the joint latent space. This approach is scalable as it only uses $M$ unimodal VAEs and one score model, and it performs better as we condition it on more modalities. That is, as the number of information available increases, the more accurate marginal distribution we get, thus increasing the generative quality. Unconditional generation from the joint distribution can also be done by sampling from the score model respecting the joint distribution. Figure 1 describes the overall architecture of the model.

Our contributions include 1) we propose a novel generative multimodal autoencoder approach that satisfies most of the appealing properties of multimodal VAEs, supported using extensive experimental studies. 2) We introduce coherence guidance to improve the coherence of observed and unobserved modalities. 3) We show that our model provides a valid variational lower bound on data likelihood.

## 2 METHODOLOGY

Assuming the each data point consists of $M$ modalities: $\mathbf{x}_{1:M} = (\mathbf{x}_1, \mathbf{x}_2, \cdots, \mathbf{x}_M)$, our latent variable model describes the data distribution as $p(\mathbf{x}_{1:M}) = \sum_{\mathbf{z}_{1:M}} p(\mathbf{x}_{1:M}|\mathbf{z}_{1:M})p(\mathbf{z}_{1:M})$, where $\mathbf{z}_{1:M} = (\mathbf{z}_1, \mathbf{z}_2, \cdots, \mathbf{z}_M)$ and $\mathbf{z}_k$ is the latent vector corresponding to the $k$th modality. In contrast to common multimodal VAE setups (Wu & Goodman, 2018; Shi et al., 2019), we don't consider any shared latent representation among different modalities, and we assume each latent variable[1] $\mathbf{z}_k$ only captures the modality-specific representation of the corresponding modality $k$. Therefore, the

---

[1]For simplicity we use "variable" to refer to the group of variables that describe the latent representation of a modality

variational lower bound on $\log p(\mathbf{x}_{1:M})$ can be written as:

$$\log p(\mathbf{x}_{1:M}) \geq \mathbb{E}_{q(\mathbf{z}_{1:M}|\mathbf{x}_{1:M})} \log \frac{p(\mathbf{x}_{1:M}|\mathbf{z}_{1:M})p(\mathbf{z}_{1:M})}{q(\mathbf{z}_{1:M}|\mathbf{x}_{1:M})}. \tag{1}$$

To simplify the joint generative model $p(\mathbf{x}_{1:M}|\mathbf{z}_{1:M})$ and the joint recognition model $q(\mathbf{z}_{1:M}|\mathbf{x}_{1:M})$, we assume two conditional indenpendencies: 1) Given an observed modality $\mathbf{x}_k$, its corresponding latent variable $\mathbf{z}_k$ is independent of other latent variables, i.e., $\mathbf{x}_k$ have enough information to describe $\mathbf{z}_k$: $\mathbf{z}_k \perp \mathbf{z}_{-k}|\mathbf{x}_k$. 2) Knowing the latent variable $\mathbf{z}_k$ of $k$th modality is enough to reconstruct that modality: $\mathbf{x}_k \perp \mathbf{x}_{-k}, \mathbf{z}_{-k}|\mathbf{z}_k$. Using these conditional independencies the generative and recognition models factorize as:

$$p(\mathbf{x}_{1:M}|\mathbf{z}_{1:M}) = \prod_{k=1}^{M} p(\mathbf{x}_k|\mathbf{z}_k) \text{ and } q(\mathbf{z}_{1:M}|\mathbf{x}_{1:M}) = \prod_{k=1}^{M} q(\mathbf{z}_k|\mathbf{x}_k).$$

Therefore we can rewrite the variational lower bound as:

$$\log p(\mathbf{x}_{1:M}) \geq \sum_k \mathbb{E}_{q_{\phi_k}(\mathbf{z}_k|\mathbf{x}_k)} \log p_{\psi_k}(\mathbf{x}_k|\mathbf{z}_k) - D_{\mathrm{KL}}(\prod_k q_{\phi_k}(\mathbf{z}_k|\mathbf{x}_k)||p_\theta(\mathbf{z}_{1:M})), \tag{2}$$

where $\phi_k$ and $\psi_k$ are the parameterizations of the recognition and generative models, respectively, and $\theta$ parameterizes the prior. If we assume the prior $p_\theta(\mathbf{z}_{1:M})$ factorizes as $\prod_k p_{\theta_k}(\mathbf{z}_k)$, then the variational lower bound of $\log p(\mathbf{x}_{1:M})$ becomes $\sum_k \mathrm{ELBO}_k$, where $\mathrm{ELBO}_k$ is the variational lower bound of individual modality. However, such an assumption ignores the dependencies among latent variables and results in a lack of coherence among generated modalities when using prior for generating multimodal samples. To benefit from the decomposable ELBO but also benefit from the joint prior, we separate the training into two steps. In step I, we maximize the ELBO with respect to $\phi$ and $\psi$ assuming prior $p(\mathbf{z}_{1:M}) = \prod \mathcal{N}(\mathbf{0}, \sigma\mathbf{I})$, which only regularizes the recognition models. In step II, we optimize ELBO with respect to $\theta$ assuming a joint prior over latent variables. Therefore, in step II maximizing the ELBO reduces to: $\min_\theta D_{\mathrm{KL}}(\prod_k q_{\phi_k}(\mathbf{z}_k|\mathbf{x}_k)||p_\theta(\mathbf{z}_{1:M}))$ and since recognition model is constant w.r.t. $\theta$, the step II becomes $\max_\theta \mathbb{E}_{p(\mathbf{x}_{1:M})}\mathbb{E}_{\prod_k q(\mathbf{z}_k|\mathbf{x}_k)} \log p_\theta(\mathbf{z}_{1:M})$, which is equivalent to maximum likelihood training of the parametric prior using sampled latent variables for each data points. And during inference, we only need samples from $p_\theta(\mathbf{z}_{1:M})$, thus we can parameterize $s_\theta(\mathbf{z}_{1:M}) \approx \nabla_{\mathbf{z}_{1:M}} \log(p_\theta(\mathbf{z}_{1:M}))$ as the score model and train $s_\theta(\mathbf{z}_{1:M})$ using score matching (Hyvärinen & Dayan, 2005). In practice, the score matching objective does not scale to the large dimension which is required in our setup, and other alternatives such as denoising score-matching (Vincent, 2011) and sliced score-matching (Song et al., 2020a) have been proposed.

Here we use denoising score matching in the continuous form that diffuses the latent samples into noise distribution and can be written in a stochastic differential equation (SDE) form (Song et al., 2020b). The forward process will be an SDE of $d\mathbf{z} = f(\mathbf{z},t)dt + g(t)d\mathbf{w}$ where $\mathbf{w}$ is the Brownian motion. To reverse the noise distribution back to the latent distribution, we use the trained score model and sample iteratively from the reverse SDE $d\mathbf{z} = \left[\mathbf{f}(\mathbf{z},t) - g(t)^2\nabla_\mathbf{z} \log p_t(\mathbf{z})\right] dt + g(t)d\overline{\mathbf{w}}$. The score network which approximates $\nabla_\mathbf{z} \log p_t(\mathbf{z})$ is trained using eq. 3:

$$\boldsymbol{\theta}^* = \arg\min_{\boldsymbol{\theta}} \mathbb{E}_t \left\{ \lambda(t) \mathbb{E}_{\mathbf{z}(0)} \mathbb{E}_{\mathbf{z}(t)|\mathbf{z}(0)} \left[ \left\| \mathbf{s}_{\boldsymbol{\theta}}(\mathbf{z}(t), t) - \nabla_{\mathbf{z}(t)} \log p_{0t}(\mathbf{z}(t) \mid \mathbf{z}(0)) \right\|_2^2 \right] \right\} \tag{3}$$

## 2.1 INFERENCE WITH MISSING MODALITIES

The goal of inference is to sample unobserved modalities (indexed by $\mathbf{u} \subseteq \{1, \cdots, M\}$) given the observed modalities (indexed by $\mathbf{o} \subset \{1, \cdots, M\}$) from $p(\mathbf{x}_\mathbf{u}|\mathbf{x}_\mathbf{o})$. We define a variational lower bound on log-probably $\log p(\mathbf{x}_\mathbf{u}|\mathbf{x}_\mathbf{o})$ using posterior $q(\mathbf{z}_\mathbf{u}|\mathbf{x}_\mathbf{o})$ on the latent variables of the unobserved modalities $\mathbf{z}_\mathbf{u}$:

$$\begin{aligned}
\log p(\mathbf{x}_\mathbf{u}|\mathbf{x}_\mathbf{o}) &= \log \mathbb{E}_{q(\mathbf{z}_\mathbf{u}|\mathbf{x}_\mathbf{o})} p(\mathbf{x}_\mathbf{u}|\mathbf{x}_\mathbf{o}, \mathbf{z}_\mathbf{u}) \\
&\geq \mathbb{E}_{q(\mathbf{z}_\mathbf{u}|\mathbf{x}_\mathbf{o})} \log p(\mathbf{x}_\mathbf{u}|\mathbf{x}_\mathbf{o}, \mathbf{z}_\mathbf{u}) \\
&= \mathbb{E}_{q(\mathbf{z}_\mathbf{u}|\mathbf{x}_\mathbf{o})} \log p(\mathbf{x}_\mathbf{u}|\mathbf{z}_\mathbf{u}),
\end{aligned} \tag{4}$$

where $p(\mathbf{x_u}|\mathbf{z_u}) = \prod_{k\in\mathbf{u}} p_{\psi_k}(\mathbf{x}_k|\mathbf{z}_k)$ and $p_{\psi_k}(\mathbf{x}_k|\mathbf{z}_k)$ is the generative model for modality $k$. We define the posterior distribution $q(\mathbf{z_u}|\mathbf{x_o})$ as the following:

$$q(\mathbf{z_u}|\mathbf{x_o}) = \sum_{\mathbf{z_o}} \left[\prod_{i\in\mathbf{o}} q_{\phi_i}(\mathbf{z}_i|\mathbf{x}_i)\right] p_\theta(\mathbf{z_u}|\mathbf{z_o}), \tag{5}$$

where $q_{\phi_k}(\mathbf{z}_k|\mathbf{x}_k)$ is the recognition model for modality $k$.

In order to sample from $q(\mathbf{z_u}|\mathbf{x_o})$, following eq. 5, we first sample $\mathbf{z_o}$ for all observed modalities, and then sample $\mathbf{z_u}$ from $p_\theta(\mathbf{z_u}|\mathbf{z_o})$ by fixing $\mathbf{z_o}$ which is the latent representation of the observed modalities and updating the unobserved ones during sampling. When all modalities are missing, we will update all modalities in the sampling step.

After running the sampling for $T$ steps we use the generative models $p(\mathbf{x_u}|\mathbf{z_u}) = \prod_{k\in\mathbf{u}} p_{\psi_k}(\mathbf{x}_k|\mathbf{z}_k)$ to sample the unobserved modalities. We use the Predictor-Corrector (PC) sampling algorithm (Song et al., 2020b) which is a mix of Euler-Maruyama and Langevin Dynamics (Welling & Teh, 2011) to sample from the score model.

## 2.2 Coherence guidance

The score-based model improves the coherence among predicted modalities, however, when the number of unobserved modalities increases, it is more likely that the predicted modalities are not aligned with the observed modalities. To address this issue, we use extra conditional guidance during the reverse process for sampling (Dhariwal & Nichol, 2021). In particular, we train an energy-based model that assigns low energy to a coherent pair of modalities and high energy to incoherent ones. We define the energy-based model $E_\omega(\mathbf{z}_o, \mathbf{z}_u)$ over the latent representations of an observed modality $\mathbf{z}_o$ and an unobserved modality $\mathbf{z}_u$. Here we assume that the latent representation of all modalities has the same dimension. During training, we randomly select two modalities $(\mathbf{x}_o, \mathbf{x}_u)$ as a positive pair and substitute the second modality with an incoherent example $\mathbf{x}_n$ in training data to construct a negative pair $(\mathbf{x}_o, \mathbf{x}_n)$. We train $E_\omega$ using noise contrastive estimation (Gutmann & Hyvärinen, 2010):

$$\max_\omega \; \mathbb{E}_{(x_o,x_u)\sim p(x_o,x_u),z_o\sim q(z|x_o),z_u\sim q(z|x_u)} \log \frac{1}{1+\exp\left(E_\omega\left((z_o,z_u)\right)\right)} \tag{6}$$

$$+ \mathbb{E}_{x_o\sim p(x_o),x_n\sim p(x_n),z_o\sim q(z|x_o),z_n\sim q(z|x_n)} \log \frac{1}{1+\exp\left(-E_\omega(z_o,z_n)\right)}$$

We also perturb the latent representations with the same perturbation kernel used to train the score model. The final score function with coherence guidance has the following form:

$$\tilde{s}(z_u) = s_\theta(z_u) - \gamma \nabla_{z_u} E_\omega(z_o, z_u), \tag{7}$$

where $\gamma$ controls the strength of guidance. During each step of inference, we randomly select $x_o$ from observed modalities and randomly select $x_u$ from unobserved modalities and update the score of $z_u$ using eq. 7.

## 3 Related Works

Our work is heavily inspired by earlier works on deep multimodal learning (Ngiam et al., 2011; Srivastava & Salakhutdinov, 2014). Ngiam et al. (2011) use a set of autoencoders for each modality and a shared representation across different modalities and trained the parameters to reconstruct the missing modalities given the present one. Srivastava & Salakhutdinov (2014) define deep Boltzmann machine to represent multimodal data with modality-specific hidden layers followed by shared hidden layers across multiple modalities, and use Gibbs sampling to recover missing modalities.

Suzuki et al. (2017) approached this problem by maximizing the joint ELBO and additional KL terms between the posterior of the joint and the individuals to handle missing data. Tsai et al. (2019) propose a factorized model in a supervised setting over model-specific representation and label representation, which capture the shared information. The proposed factorization is $q(\mathbf{z}_{1:M}, \mathbf{z}_y|\mathbf{x}_{1:M}, \mathbf{y}) = q(\mathbf{z}_y|\mathbf{x}_{1:M}) \prod_{k=1}^M q(\mathbf{z}_k|\mathbf{x}_k)$, where $\mathbf{z}_y$ is the latent variable corresponding to the label.

Most current approaches define a multimodal variational lower bound similar to variational autoencoders Kingma & Welling (2014) using a shared latent representation for all modalities:

$$\log p(\mathbf{x}_{1:M}) \geq \mathbb{E}_{q_\phi(\mathbf{z}|\mathbf{x}_{1:M})} \left[ \log p_\psi(\mathbf{x}_{1:M}|z) \right] - D_{\mathrm{KL}}(q_\phi(z|\mathbf{x}_{1:M}) \| p(\mathbf{z})). \tag{8}$$

Similar to our setup $p_\psi(\mathbf{x}_{1:M}|z) = \prod_k p_{\psi_k}(\mathbf{x}_k|\mathbf{z})$, but $q_\phi(\mathbf{z}|\mathbf{x}_{1:M})$ is handled differently. Wu & Goodman (2018) use a product of experts to describe $q$: $q_{\mathrm{PoE}}(\mathbf{z}|\mathbf{x}_{1:M}) = p(\mathbf{z}) \prod_{k=1}^M q_{\phi_k}(\mathbf{z}|\mathbf{x}_k)$. Assuming $p(\mathbf{z})$ and each of $q_{\phi_k}(\mathbf{z}|\mathbf{x}_k)$ follow a Gaussian distribution, $q_{\mathrm{PoE}}$ can be calculated in a closed form, and we can optimize the multimodal ELBO accordingly. To get a good performance on generating missing modality, Wu & Goodman (2018) sample different ELBO combinations of the subset of modalities. Moreover, the sub-sampling proposed by Wu & Goodman (2018) results in an invalid multimodal ELBO (Wu & Goodman, 2019). The MVAE proposed by (Wu & Goodman, 2018) generates good-quality images but suffers from low cross-modal coherence. To address this issue Shi et al. (2019) propose constructing $q_\phi(\mathbf{z}|\mathbf{x}_{1:M})$ as a mixture of experts: $q_{\mathrm{MoE}}(\mathbf{z}|\mathbf{x}_{1:M}) = \frac{1}{M} \sum_k q_{\phi_k}(\mathbf{z}|\mathbf{x}_k)$. However, as pointed out by Daunhawer et al. (2022), sub-sampling from the mixture component results in lower generation quality, while improving the coherence.

Sutter et al. (2021) propose a mixture of the product of experts for $q$ by combining these two approaches: $q_{\mathrm{MoPoE}}(\mathbf{z}|\mathbf{x}_{1:M}) = \frac{1}{2^M} \sum_{\mathbf{x_m}} q(\mathbf{z}|\mathbf{x_m})$, where $q(\mathbf{z}|\mathbf{x_m}) = \prod_{k \in \mathbf{m}} q(\mathbf{z}|\mathbf{x}_k)$ and $\mathbf{m}$ is a subset of modalities. The number of mixture components grows exponentially as the number of modalities increases. MoPoE has better coherence than PoE, but as discussed by Daunhawer et al. (2022) sub-sampling modalities in mixture-based multimodal VAEs result in loss of generation quality of the individual modalities. To address this issue, more recently and in parallel to our work, Palumbo et al. (2023) introduce modality-specific latent variables in addition to the shared latent variable. In this setting the joint probability model over all variables factorizes as $p(\mathbf{x}_{1:M}, \mathbf{z}, \mathbf{w}_{1:M}) = p(\mathbf{z}) \prod_k p(\mathbf{x}_k|\mathbf{z}, \mathbf{w}_k) p(\mathbf{w}_k)$ and $q$ factorizes as $q_{\mathrm{MMVAE+}}(\mathbf{z}, \mathbf{w}_{1:M}|\mathbf{x}_{1:M}) = q_{\phi_z}(\mathbf{z}|\mathbf{x}) \prod_k q_{\phi_k}(\mathbf{w}_k|\mathbf{x}_k)$. Using modality-specific representation is also explored by Lee & Pavlovic (2021), however, the approach proposed by Palumbo et al. (2023) is more robust to controlling modality-specific representation vs shared representation. But since the shared component is a mixture of experts of individual components, it can only use one of them at a time during inference which limits its ability to use additional observations that are available as the number of given modalities increase.

Sutter et al. (2020) propose an updated multimodal objective that consists of a JS-divergence term instead of the normal KL term with a mixture-of-experts posterior. They also add additional modality-specific terms and a dynamic prior to approximate the unimodal and the posterior term. Though these additions provide some improvement, there is still a need for a model that balances coherence with quality (Palumbo et al., 2023).

Wolff et al. (2022) propose a hierarchical multimodal VAEs for the task where a generative model of the form $p(x, g, z)$ and an inference model $q(g, z|x_{1:M})$ containing multiple hierarchies of $z$ where $g$ holds the shared structures of multiple modalities in a mixture-of-experts form $q(g|x_{1:M})$. They argue that the modality-exclusive hierarchical structure helps in avoiding the modality sub-sampling issue and can capture the variations of each modality. Though the hierarchy gives some improvement in results, the model is still restricted in capturing the shared structure in $g$ discussed in their work.

Suzuki & Matsuo (2023) introduce a multimodal objective that avoids sub-sampling during training by not using a mixture-of-experts posterior to avoid the main issue discussed by Daunhawer et al. (2022). They propose a product-of-experts posterior multimodal objective with additional unimodal reconstruction terms to facilitate cross-modal generations.

Hwang et al. (2021) propose an ELBO that is derived from an information theory perspective that encourages finding a representation that decreases the total correlation. They propose another ELBO objective which is a convex combination of two ELBO terms that are based on conditional VIB and VIB. The VIB term decomposes to ELBOs of previous approaches. The conditional term decreases the KL between the joint posterior and individual posteriors. They use a product-of-experts as their joint posterior.

Finally, parallel to our work, Xu et al. (2023) propose diffusion models to tackle the multimodal generation which uses multi-flow diffusion with data and context layer for each modality and a shared global layer.

## 4 EXPERIMENTS

We run experiments on an extended version of PolyMNIST (Sutter et al., 2021) and high-dimensional CelebAMask-HQ (Lee et al., 2020) datasets.

### 4.1 EXTENTED POLYMNIST

The original PolyMNIST introduced by Sutter et al. (2021) has five modalities and in order to study the behavior of the methods on a larger number of modalities, we extended the number of modalities to ten. Figure 2 shows samples of the Extended PolyMNIST data.

We compare our methods SBM-VAE and SBM-RAE, which substitute individual variational autoencoder with a regularized deterministic autoencoder (Ghosh et al., 2020); see Appendix A.1 for the details of SBM-RAE, as well as their counterparts with coherence guidance SBM-VAE-C and SBM-RAE-C with MVAE (Wu & Goodman, 2018), MMVAE (Shi et al., 2019), MoPoE (Sutter et al., 2021), MVTCAE (Hwang et al., 2021), and MMVAE+ (Palumbo et al., 2023).

We use the same residual encoder-decoder architecture similar to (Daniel & Tamar, 2021) for all methods except for MMVAE+ which performs poorly on this architecture. Therefore, we use the original architecture described in Palumbo et al. (2023) for MMVAE+. See Appendix A.2 for details of the used neural networks.

We use $\beta$-VAE for training baseline multimodal VAEs and unimodal VAEs with Adam optimizer (Kingma & Ba, 2015). We use an initial learning rate of 0.001, and the models are trained for 200 epochs with different $\beta$ values and the best model is selected using the validation set for each baseline. The unimodal VAEs that are later used for SBM-VAE and SBM-RAE use a latent dimension of 64 which is reshaped to 8x8. MMVAE+ also uses a total of 64 dimension latent vector. We increased the latent dimension of the other baselines to $64 * n$ where $n$ is the number of modalities the model uses to compensate the fact that they

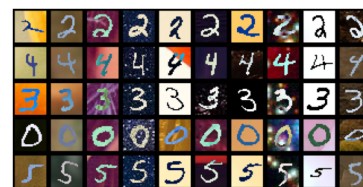

Figure 2: Updated PolyMnist Dataset

don't have modality specific representations. Other hyperparameters are discussed in Appendix A.2.

The score networks are trained with an initial learning rate of $0.0002$ using the Adam optimizer. We use the variance preserving SDE (VPSDE) as the perturbation kernel. We run PC (Predictor-Corrector) sampling technique using 1 step Langevin Corrector for $N = 100$ number of steps to generate samples from the score models. For the score models, $\mathbf{z_u}$, missing modalities, are initialized from a Gaussian distribution of $\mathcal{N}(\mathbf{0}, \mathbf{1})$. See Appendix A.3 for the details of training and sampling using the score model.

Conditional generation is done for MVAE and MVTCAE using PoE of the posteriors of the given modalities, and for the mixture models, a mixture component is chosen uniformly from the observed modalities.

We evaluate all methods on both prediction coherence and generative quality. To measure the coherence, we use a pre-trained classifier to extract the label of the generated output and compare it with the associated label of the observed modalities Shi et al. (2019). The coherence of the unconditional generation is evaluated by counting the number of consistent predicted labels from the pre-defined classifier. We also measure the generative quality of the generated modalities using the FID score (Heusel et al., 2017).

Figure 3 shows the generated samples from the third modality given the rest. The SBM-VAE generates high-quality images with considerable variation, very close to the training dataset, while the baselines generate more blurry images with a lower variation. We also study, how the conditional coherence and FID change with the number of the observed modality. For an accurate conditional model, we expect as we observe more modalities, the accuracy of conditional generation improves without decreasing the quality. Figure 4 shows the FID score of the last modality as we increase the given modalities and Figure 5 demonstrates the conditional coherence by measuring the accuracy of the predicted images on the last modality given a different number of observed modalities.

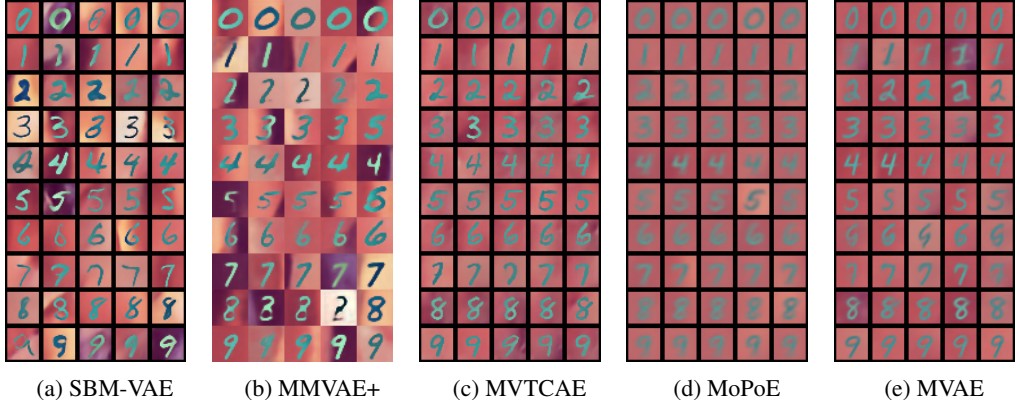

| (a) SBM-VAE | (b) MMVAE+ | (c) MVTCAE | (d) MoPoE | (e) MVAE |

Figure 3: Multiple conditionally generated samples for each digit from the third modality. Each column shows samples, from 0 to 9, generated conditionally given the remaining modalities. (Please see the appendix for the complete set that includes SBM-RAE and MMVAE.)

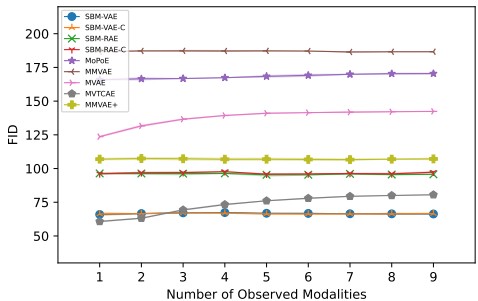

Figure 4: The conditional FID of the last modality generated by incrementing the given modality at a time. The x-axis shows how many modalities are given to generate the modality and the y-axis shows the FID score of the generated modality.

Figure 5: The conditional accuracy of the last modality generated by incrementing the given modality at a time. The x-axis shows how many modalities are given to generate the modality and the y-axis shows the accuracy of the generated modality.

As mentioned by Daunhawer et al. (2022), the other multimodal VAEs' generative qualities decline as we observe more modalities. This is shown by the high FID score of the generated modality for the compared methods except for the SBM models and MMVAE+. SBM-based multimodal VAEs and MMVAE+ both are equipped with a way to capture modality-specific representation, so increasing the number of observed modalities does not affect their generation quality. The SBM exploits the additional modalities to generate better and more accurate images as we increase the number of modalities without losing quality. Even though the other multimodal VAEs have high coherence as expected, however, as we increase the number of observed modalities the SBM-based models achieve high coherence too. Similar experiments have been reported for predicting the first modality in Appendix (see Figure 11 and Figure 12).

The plain score model suffers from low coherence when we have only one or two observed modalities. But with the help of the coherence guidance, we can guide the model towards the correct output and that helps increase the accuracy of SBM-VAE and SBM-RAE. In addition to that, increasing the number of sampling steps from 100 to 1000 also helps when a few modalities are observed. We demonstrate the effect of coherence guidance and the number of inference steps in Figure 7.

We also study unconditional coherence, which has been reported in Figure 6. We evaluate unconditional accuracy by counting the number of consistent labels across different modalities after classifying the generated images using a pre-trained classifier. 71% of generated images using SBM

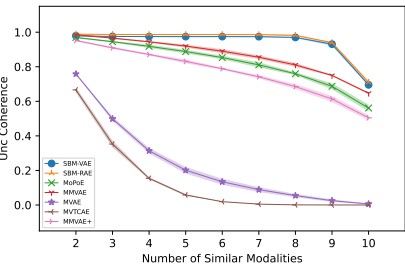

Figure 6: Unconditional coherence. The x-axis shows the number of coherent modalities and the y-axis shows the percentage of such coherent predicted modalities in the generated output

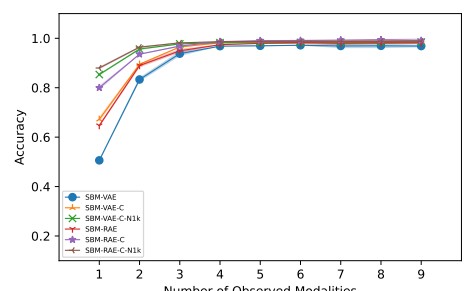

Figure 7: Effect of having the energy-based coherence guidance model and number of sampling steps

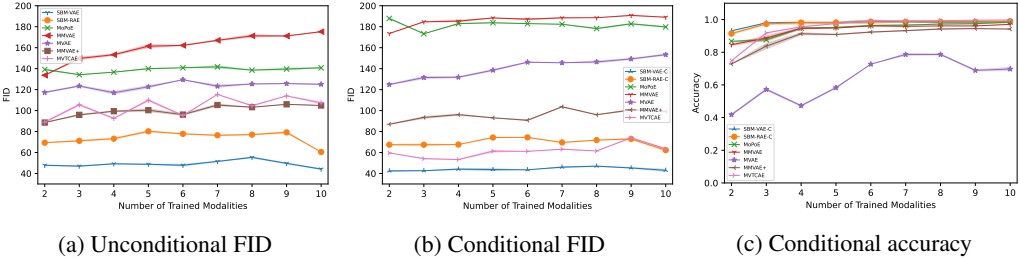

(a) Unconditional FID        (b) Conditional FID        (c) Conditional accuracy

Figure 8: a) unconditional FID (b) conditional FID and c) conditional accuracy of the generated first modality given the rest of modalities by increasing the number of trained modalities.

models are coherent which are considerably better than the other baselines. Our result shows that MVTCAE performs poorly in unconditional generation.

Finally, we study the scalability of methods given a different number of data modalities (see Figure 8). We first train the models with different numbers modalities and then evaluate unconditional FID for the first modality (for unconditional FID we don't use coherence guidance since there is no observed modality), conditional FID for the first modality given the rest, and conditional accuracy of the first modality given the rest. The performance of our proposed methods is consistent and superior across different number of data modalities. Specifically, both SBM-VAE-C and SBM-RAE-C show consistent FID and coherence, for which the complexity of the sampling from prior remains the same as we increase the number of data modalities. Further details as well as more generated samples have been reported in Appendix A.4.

## 4.2 CELEBAMASK-HQ

The images, masks, and attributes of the CelebAMask-HQ dataset can be treated as different modalities of expressing the visual characteristics of a person. A sample from the dataset is shown in Figure 9. The images and masks are resized to 128 by 128. The masks are either white or black where all the masks given in the CelebAMask-HQ except the skin mask are drawn on top of each other as one image. We follow the pre-processing of Wu & Goodman (2018) by selecting 18 out of 40 existing attributes as described. We compare SBM-VAE, SBM-VAE-C, SBM-RAE, SBM-RAE-C, MoPoE, MVT-CAE, and MMVAE+. See Appendix A.6 for the experimental setups of these methods.

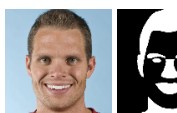

Figure 9: A sample of image and associated mask from the CelebAMask-HQ dataset (Attribute not shown)

We evaluate the generation quality of the image modality using FID score and generation accuracy of mask and attribute modalities using sample-average $F_1$ score. Table 1 shows how our methods

compare with the baselines in the presence of zero (unconditional), one, or two observed modalities. We have also reported the performance of a supervised model trained to predict attributes and masks directly given the images. SBM-VAE (and SBM-VAE-C) generates high-quality images compared to baselines, in both conditional and unconditional settings, while on the mask and attribute modalities, MoPoE and MVTCAE achieve a better $F_1$ score. This is consistent with the capacity of MoPoE and MVTCAE for joint representation which is more useful for MAP prediction. We can also observe that conditional guidance consistently improves the coherence of score-based models.

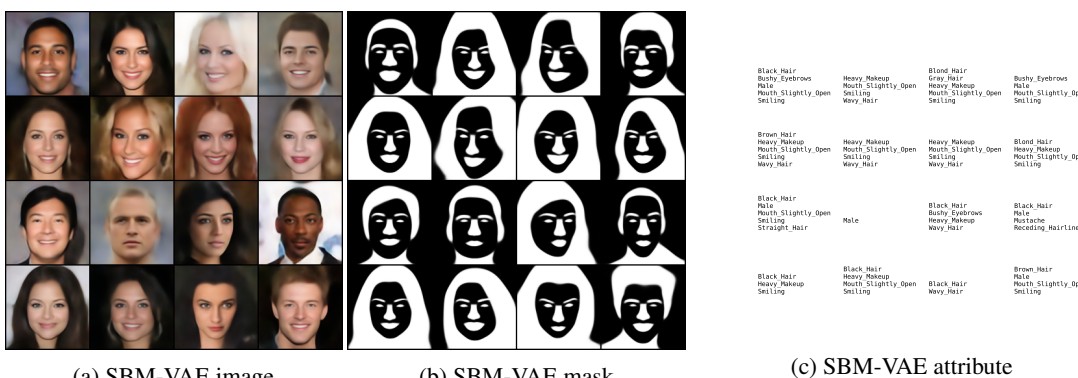

(a) SBM-VAE image  (b) SBM-VAE mask  (c) SBM-VAE attribute

Figure 10: Unconditional generation using SBM-VAE

Table 1: CelebAMask-HQ Result

| GIVEN | Attribute | | Image | | | | Mask | |
| --- | --- | --- | --- | --- | --- | --- | --- | --- |
| | Both | Img | Both | Mask | Attr | Unc | Both | Img |
| | F1 | F1 | FID | FID | FID | FID | F1 | F1 |
| SBM-RAE | $0.62_{(\pm0.03)}$ | $0.6_{(\pm0.004)}$ | $84.9_{(\pm0.19)}$ | $86.4_{(\pm0.02)}$ | $85.6_{(\pm0.48)}$ | $84.2_{(\pm0.25)}$ | $0.83_{(\pm0.001)}$ | $0.82_{(\pm0.001)}$ |
| SBM-RAE-C | $0.66_{(\pm0.03)}$ | $0.64_{(\pm0.004)}$ | $83.6_{(\pm0.05)}$ | $82.8_{(\pm0.05)}$ | $83.1_{(\pm0.14)}$ | $84.2_{(\pm0.05)}$ | $0.83_{(\pm0.004)}$ | $0.82_{(\pm0.004)}$ |
| SBM-VAE | $0.62_{(\pm0.003)}$ | $0.58_{(\pm0.005)}$ | $\mathbf{81.6}_{(\pm0.10)}$ | $81.9_{(\pm0.03)}$ | $78.7_{(\pm0.25)}$ | $\mathbf{79.1}_{(\pm0.07)}$ | $0.83_{(\pm0.001)}$ | $0.83_{(\pm0.001)}$ |
| SBM-VAE-C | $0.69_{(\pm0.005)}$ | $0.66_{(\pm0.001)}$ | $82.4_{(\pm0.1)}$ | $\mathbf{81.7}_{(\pm0.29)}$ | $\mathbf{76.3}_{(\pm0.7)}$ | $\mathbf{79.1}_{(\pm0.3)}$ | $0.84_{(\pm0.02)}$ | $0.84_{(\pm0.001)}$ |
| MoPoE | $0.68_{(\pm0.002)}$ | $\mathbf{0.71}_{(\pm0.004)}$ | $114.9_{(\pm0.32)}$ | $101.1_{(\pm0.16)}$ | $186.7_{(\pm0.28)}$ | $164.8_{(\pm0.62)}$ | $0.85_{(\pm0.002)}$ | $\mathbf{0.92}_{(\pm0.001)}$ |
| MVTCAE | $\mathbf{0.71}_{(\pm0.001)}$ | $0.69_{(\pm0.004)}$ | $94_{(\pm0.45)}$ | $84.2_{(\pm0.32)}$ | $87.2_{(\pm0.08)}$ | $162.2_{(\pm1.08)}$ | $\mathbf{0.89}_{(\pm0.001)}$ | $0.89_{(\pm0.003)}$ |
| MMVAE+ | $0.64_{(\pm0.003)}$ | $0.61_{(\pm0.002)}$ | $133_{(\pm14.28)}$ | $97.3_{(\pm0.04)}$ | $153_{(\pm0.49)}$ | $103.7_{(\pm0.61)}$ | $0.82_{(\pm0.03)}$ | $0.89_{(\pm0.002)}$ |
| Supervised | | $0.79_{(\pm0.001)}$ | | | | | | $0.94_{(\pm0.001)}$ |

Finally, we show the unconditional generation across different modalities using SBM-VAE in Figure 10. Please see Appendix A.7 for more qualitative samples on the CelebAMask-HQ dataset.

## 5 CONCLUSION AND DISCUSSION

Multimodal VAEs are an important tool for modeling multimodal data. In this paper, we provide a different multimodal posterior using score-based models. Our proposed method learns the correlation of latent spaces of unimodal representations using a joint score model in contrast to the traditional multimodal VAE ELBO. We show that our method can generate better-quality samples and at the same time preserves coherence among modalities. We have also shown that our approach is scalable to multiple modalities. We also introduce coherence guidance to improve the coherence of the generated modalities with the observed modalities. The coherence guidance effectively improves the coherence of our score-based models. Our proposed methods assume the presence of all modalities during training which is a limitation of the current work, and in future research, we are going to address this issue. Moreover, compared to traditional multimodal VAEs, score-based multimodal VAEs require an expensive sampling procedure during inference, which creates a trade-off between the quality and coherence of the prediction and the computational cost.

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

## A    APPENDIX

### A.1    SBM-RAE

Regularized autoencoders (RAEs) (Ghosh et al., 2020) can be used instead of VAEs in our setting. RAEs assume a deterministic encoder and regularize the latent space directly by penalizing $L_2$ norm of the latent variables:

$$\mathcal{L}_{\text{RAE}} = \mathcal{L}_{\text{REC}} + \beta||\mathbf{z}||_2^2 + \lambda \mathcal{L}_{\text{REG}}, \tag{9}$$

where $L_{\text{REC}}$ is the reconstruction loss for deterministic autoencoder and $L_{\text{REG}}$ is the decoder regularizer. In our setup, we don't use any decoder regularizer.

In order to generate a sample from the latent space, RAEs require to fit separate density estimators on the latent variables. In our case, the score models are responsible for generating samples from the latent space, which makes RAE a compelling choice for our setup. RAEs are capable of learning more complex latent structures, and expressive generative models such as score models can effectively learn that structure to generate high-quality samples.

### A.2    MODEL ARCHITECTURES AND EXPERIMENTAL SETUPS FOR EXTENDED POLYMNIST

The extended PolyMnist dataset was updated from the original PolyMnist dataset by Sutter et al. (2020) with different background images and ten modalities. It has 50,000 training set, 10,000 validation set, and 10,000 test set. The VAEs for each modality are trained with an initial learning rate of 0.001 using a $\beta$ value of 0.1 where all the prior, posterior, and likelihood are Gaussians. This also applies to all multimodal VAEs except MMVAE+ which uses Laplace distribution instead of Gaussian. The RAE for each modality was trained using the mean squared error loss with the norm of $||\mathbf{z}||_2^2$ regularized by a factor of $10^{-5}$ and a Gaussian noise added to $z$ before feeding to the decoder with mean 0 and variance of 0.01 where the hyperparameter values were tuned using the validation set. The encoders and decoders for all models use residual connections to improve performance and are similar in structure to the architecture used in Daniel & Tamar (2021) except for MMVAE+ were we used the original model because the model's performance doesn't generalize to different neural net architecture. For MMVAE+, modality-specific and shared latent sizes are each 32 and the model was trained similarly to the code provided by the paper [2] with the IWAE estimator with K=1. The detailed architecture can be found by referring to the code that is attached. We used latent size of 64 for our models and MMVAE+ and we increased the latent dimension of the other baselines to $64 * n$ where $n$ is the number of modalities because they don't have any modality specific representation. We chose the best $\beta$ value for each model using the validation set. For MoPoE and MMVAE, $\beta$ of 2.5 was chosen, for MMVAE+ 5, for MVAE 1 and for MVTCAE 0.1.

The neural net of SBM-VAE and SBM-RAE is a UNET network where we resize the latent size to 8x8. For SBM-VAE, samples are taken from the posterior at training time and the mean of posterior is taken at inference time. For SBM-RAE, the $\mathbf{z}$s are taken directly. We use a learning rate of 0.0002 with the Adam optimizer (Kingma & Ba, 2015). The detailed hyperparameters are shown in table 2. We use the VPSDE with $\beta_0 = 0.1$ and $\beta_1$=5 with $N = 100$ and the PC sampling technique with Euler-Maruyama sampling and langevin dynamics. For modalities less than 10, we use $\beta_0$ of 1, the others hyperparameters remain the same. The energy-based model is a simple MLP with the softplus activation. We follow equation 6 to train the models.

Table 2: Score Hyperparameters PolyMnist

| Model | $\beta_{min}$ | $\beta max$ | N | LD per step | EBM-scale | batch-size |
|---|---|---|---|---|---|---|
| SBM 10mod | 0.1 | 5 | 100 | 1 | 1000 | 256 |
| SBM below 10mod | 1 | 5 | 100 | 1 | 1000 | 256 |

---

[2]MMVAE+ code is taken and updated from the official repo provided at https://github.com/epalu/mmvaeplus

### A.3 TRAINING AND INFERENCE ALGORITHM

We follow the Song et al. (2020b) to train the score models using the latent representation from the encoders. The following algorithms 1, 2 show the training and inference algorithm we use.

---
**Algorithm 1** Training

---
**Require:** $M, N,$                                             ▷ M - modality, N -epochs
  $\mathbf{z} = []$
  **for** $i = 1$ to $M$ **do**
      Get $\mathbf{x}_i$                                         ▷ Get the input x from modality i
      Sample $\mathbf{z}_i$ from $q(\mathbf{z}|\mathbf{x}_i)$                            ▷ Sample z from the encoder
      Append $\mathbf{z}_i$ to $\mathbf{z}$
  **end for**
  **for** $epoch = 1$ to $N$ **do**
      Diffuse $\mathbf{z}$
      Compute $\mathbf{s}_\theta(\mathbf{z}, t)$
      Calculate loss using equation 3 multiplied by $\sigma_t^2$
      Backpropagate and update the weights of the SBM
  **end for**

---

---
**Algorithm 2** Inference

---
  **if** Unconditional **then**
      Sample $\mathbf{z}$ from $\mathcal{N}(\mathbf{0}, \mathbf{I})$ and stack to get $\mathbf{Z}$
  **else if** Conditional **then**
      Sample $\mathbf{z}$ from $\mathcal{N}(\mathbf{0}, \mathbf{I})$ if missing
      Sample $\mathbf{z}$ from encoder $q(\mathbf{z}|\mathbf{x}_i)$ if present
      Stack all $\mathbf{z}$
  **end if**
  **for** $i = 1$ to $n$ **do**                                 ▷ n is number of sampling iterations
      Update missing $\mathbf{z}$s by the following equation using the PC sampling
      **if** use guidance **then**
          Add gradient from EBM to the score
      **end if**
  **end for**
  Feed the $\mathbf{z}$ to the respective decoder to get output

---

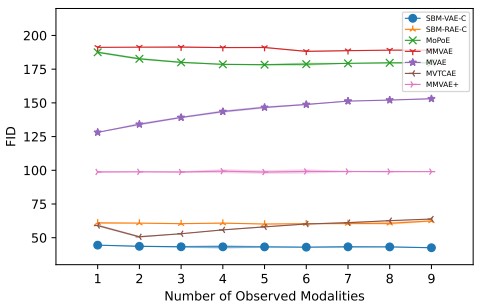
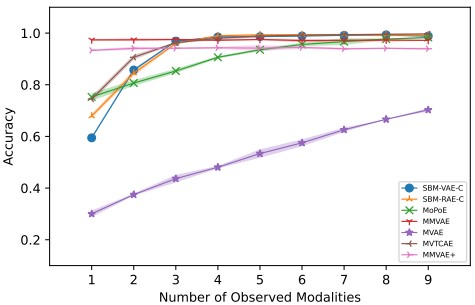

Figure 11: The conditional FID of the first modality generated by incrementing the given modality at a time. The x-axis shows how many modalities are given to generate the modality and the y-axis shows the FID score of the generated modality.

Figure 12: The conditional accuracy of the first modality generated by incrementing the given modality at a time. The x-axis shows how many modalities are given to generate the modality and the y-axis shows the accuracy of the generated modality.

| Model | Avg Coherence | Avg FID |
|---|---|---|
| MVAE | $0.751_{(\pm 0.001)}$ | $139.47_{(\pm 0.055)}$ |
| MMVAE | $0.969_{(\pm 0.055)}$ | $176.49_{(\pm 0.135)}$ |
| MoPoE | $0.982_{(\pm 0.001)}$ | $163.31_{(\pm 0.590)}$ |
| MVTCAE | $0.983_{(\pm 0.001)}$ | $77.69_{(\pm 0.200)}$ |
| MMVAE+ | $0.937_{(\pm 0.130)}$ | $110.65_{(\pm 0.0005)}$ |
| SBM-VAE | $0.967_{(\pm 0.001)}$ | $73.29_{(\pm 0.080)}$ |
| SBM-RAE | $0.968_{(\pm 0.001)}$ | $81.70_{(\pm 0.120)}$ |

Table 3: Conditional Performance

## A.4 ABLATION STUDY

### A.4.1 EXTENDED POLYMNIST RESULTS

Here we show additional graphs for different modalities in addition to the ones shown in the main paper. Figure 11 shows the same setup as figure 4 in the main paper but for another modality.

We also show the average conditional coherence and average conditional FID for each model averaging over the modalities generated given the rest. Table 3 shows the result of that. Figure 13 and 14 show the results per modality.

### A.4.2 $\beta$ ABLATION

In Table 4, we discuss how $\beta$ of the unimodal VAE affects the conditional FID and the conditional coherence of SBM-VAE. As the score model is trained on the samples from the posterior, samples that have good reconstruction are important for the score model to learn well. This is also evident in the result as lower $\beta$ values have better result than higher $\beta$ values as VAEs with lower $\beta$ values have better reconstruction. The score model also learns the gap created in unconditional sampling due to this trade-off as the score model transforms normal Gaussian noise to posterior samples during inference. The table shows a score model on the first two modalities of Extended PolyMnist trained using different $\beta$ values of 0.1,0.5, and 1 and their average conditional result.

In figure 15, 16, 18, and 15, we also show results of different $\beta$ values for MMVAE+ and MVTCAE predicting the last modality as the number of observed modalities is increased from 1 to 9.

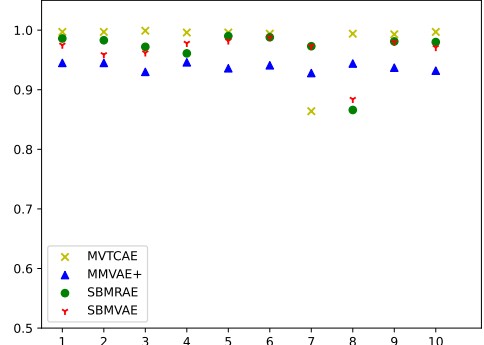

Figure 13: Conditional Accuracy (Coherence) of each modality given the rest.

Figure 14: Conditional FID of each modality given the rest.

## A.5 MODE COVERAGE

In this section, in addition to FID, we discuss an evaluation technique for unconditional generation. We generate 10,000 unconditionally generated images from the multimodal Extended PolyMnist dataset and count how many digits are generated from the 10,000 images for each digit. A good model

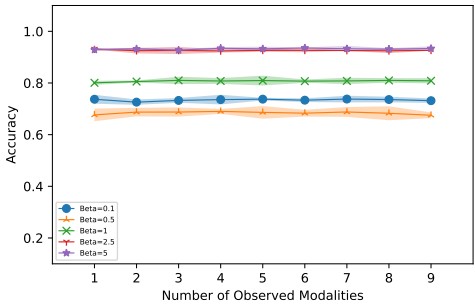

Figure 15: The conditional accuracy of the last modality generated by incrementing the given modality at a time for different $\beta$s of MMVAE+

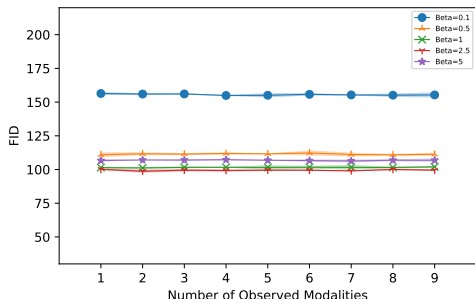

Figure 16: The conditional FID of the last modality generated by incrementing the given modality at a time for different $\beta$s of MM-VAE+.

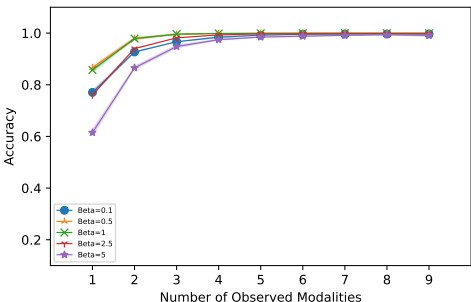

Figure 17: The conditional accuracy of the last modality generated by incrementing the given modality at a time for different $\beta$s of MVTCAE

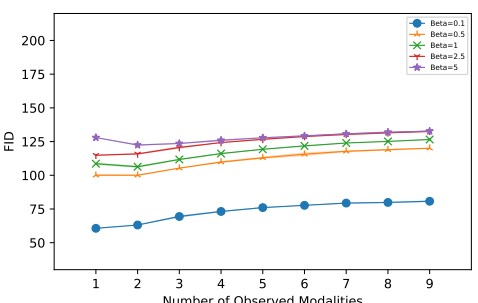

Figure 18: The conditional FID of the last modality generated by incrementing the given modality at a time for different $\beta$s of MVT-CAE.

| $\beta$ | Avg Coherence | Avg FID |
|---------|---------------|---------|
| 0.1 | $0.905_{(\pm 0.001)}$ | $52.12_{(\pm 0.045)}$ |
| 0.5 | $0.780_{(\pm 0.001)}$ | $102.85_{(\pm 0.050)}$ |
| 1 | $0.714_{(\pm 0.003)}$ | $108.35_{(\pm 0.295)}$ |

Table 4: Effect of $\beta$ on SBM-VAE

should generate all digits uniformly covering all the distribution of the dataset. Figure 19 shows the counts of each modality in a bar graph for SBMVAE, SBMRAE, MMVAE+, and MVTCAE. Our models cover and MMVAE+ cover most modes almost uniformly where as MVTCAE has some non-uniformity. Our models also have the highest unconditional coherence which means they also generate coherent outputs at the same time which is an ideal property.

### A.5.1 FINE-TUNNING THE GENERATIVE MODELS USING MISSING MODALITES

We can further finetune the generative model (decoder) to increase the overall coherence. During training, we assume all the modalities are present. This condition is necessary for training $p_\theta$ in the described setup [3]. However, we are interested in conditional queries of $p(.|\mathbf{x_o})$, we can achieve a tighter lower bound by further optimizing the $p_\psi(\mathbf{x_u}|\mathbf{z}_u)$ for sample from $q(\mathbf{z_u}|\mathbf{z_o}, \mathbf{x_o})$. We update

---

[3]We leave training $p_\theta$ with missing modalities for future work.

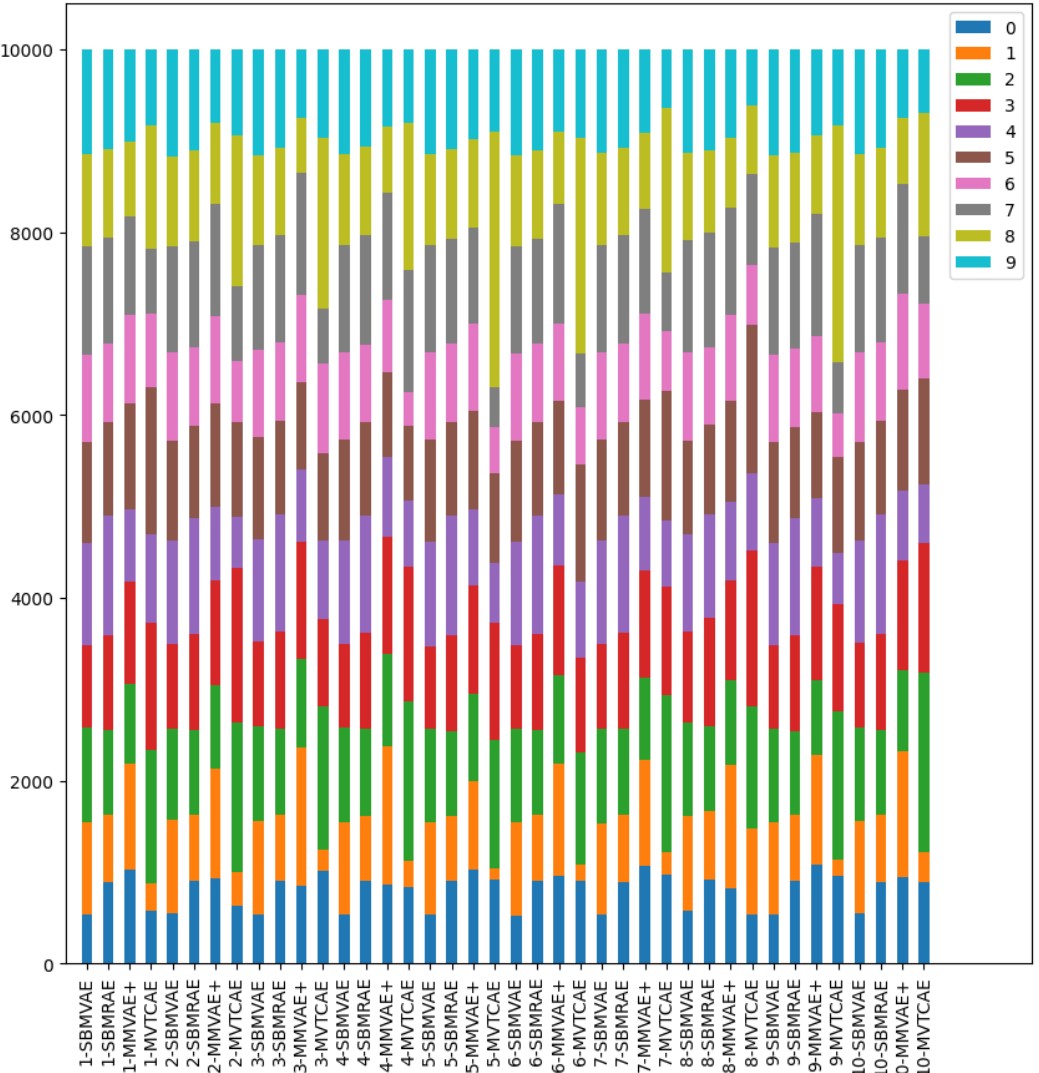

Figure 19: Mode Coverage

Table 5: Conditional Coherence with different amounts of data given [different score-net]

| Model | Drop Probability | | | | | |
|---|---|---|---|---|---|---|
| | 0.1 | 0.3 | 0.5 | 0.7 | 0.9 | Avg |
| MoPoE | 0.980 | 0.959 | 0.915 | 0.856 | 0.799 | $0.901_{(\pm0.066)}$ |
| MMVAE | 0.973 | 0.970 | 0.969 | 0.971 | 0.970 | $0.970_{(\pm0.001)}$ |
| MVAE | 0.735 | 0.638 | 0.521 | 0.430 | 0.317 | $0.528_{(\pm0.147)}$ |
| MVTCAE | 0.984 | 0.976 | 0.942 | 0.895 | 0.776 | $0.914_{(\pm0.076)}$ |
| MMVAE+ | 0.936 | 0.936 | 0.940 | 0.938 | 0.938 | $0.0.937_{(\pm0.001)}$ |
| SBMRAE | 0.949 | 0.944 | 0.896 | 0.686 | 0.425 | $0.78_{(\pm0.201)}$ |
| SBMVAE | 0.984 | 0.932 | 0.875 | 0.671 | 0.277 | $0.747_{(\pm0.258)}$ |
| SBMVAE-ft-0.1 | 0.997 | 0.980 | 0.873 | 0.600 | 0.270 | $0.744_{(\pm0.276)}$ |
| SBMVAE-ft-0.3 | 0.993 | 0.975 | 0.842 | 0.539 | 0.336 | $0.737_{(\pm0.258)}$ |
| SBMVAE-ft-0.5 | 0.996 | 0.994 | 0.915 | 0.700 | 0.260 | $0.773_{(\pm0.278)}$ |
| SBMVAE-ft-0.7 | 0.969 | 0.944 | 0.893 | 0.588 | 0.279 | $0.734_{(\pm0.265)}$ |
| SBMVAE-ft-0.9 | 0.799 | 0.802 | 0.753 | 0.534 | 0.236 | $0.624_{(\pm0.217)}$ |

the parameters of decoders to maximize the conditional log-likelihood:

$$\max_{\psi} \mathbb{E}_{q(\mathbf{z_u}|\mathbf{z_o},\mathbf{x_o})} \log p_{\psi}(\mathbf{x_u}|\mathbf{z_u})$$

$$= \max_{\psi} \frac{1}{K} \sum_{k=1}^{K} \log p_{\psi_k}(\mathbf{x_u^k}|\mathbf{z_u^k}) \quad z_{\mathbf{u}}^k \sim q(.|\mathbf{z_o},\mathbf{x_o}) \tag{10}$$

For each training example in the batch, we randomly drop each modality with probability $p$. Eq. 10 will increase the likelihood of the true assignment of the dropped modalities given the observed modalities. We evaluate this experiments on a different score model trained using NCSN Song & Ermon (2019) and with unimodal VAEs with $\beta$=0.5.

Figure 21 shows the conditional coherence of different finetuned models with varying $p$. As it can be seen in the figure, the coherence improves by a small amount but the downside of this is that it comes with worse FID values due to the fact we are only finetuning the generative model of the VAE.

Table 5 shows the conditional performance when any modality is dropped by a probability $p$ from 0.1 to 0.9. The table also includes finetuned SBMVAE (SBMVAE-ft) evaluations in addition to all models. SBMVAE-ft with drop probability 0.5 improves the overall coherence of SBMVAE by some amount.

### A.5.2 QUALITATIVE RESULTS FOR EXTENDED POLYMNIST

Here we show some conditionally generated samples and unconditional generation from each model. Conditional samples are shown in figures 23 and 24 and unconditional samples in figure 25.

### A.6 CELEBMASKHQ EXPERIMENTAL SETUP

The CelebMaskHQ dataset is taken from Lee et al. (2020) where the images, masks, and attributes are the three modalities. All face part masks were combined into a single black-and-white image except the skin mask. Out of the 40 attributes, 18 were taken from it similar to the setup of Wu & Goodman (2018). The encoder and decoder architectures are similar to Daniel & Tamar (2021) except MMVAE+ for the same reason as in PolyMNIST. A latent size of 256 was used for SBM models and MMVAE+, and a latent size of 1024 was used for MVTCAE and MoPoE. For MMVAE+, modality-specific and shared latent sizes are each 128 trained with IWAE estimator with K=1. For SBM-VAE, the image VAE was trained using Gaussian likelihood, posterior, and prior with $\beta = 0.1$. The same applies for the mask VAE but with $\beta = 1$. The attribute VAE uses Gaussian prior and posterior with a bernoulli likelihood. This applies to all other baselines with the exception of MMVAE+ which uses laplace likelihood and prior. We select the best $\beta$ for the baselines from [0.1,0.5,1,2.5,5]. MoPoE use $\beta$ of 0.1, MVTCAE 0.5 and MMVAE+ 5. For SBM-RAE, $\beta$ values of $10^{-4}$, $10^{-5}$, $10^{-4}$ and Gaussian

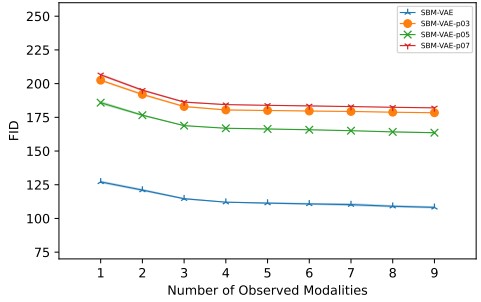

Figure 20: Plot of finetuning experiment using different $p$ values on how it affects conditional FID (different score model). The conditional FID of the last modality generated by incrementing the given modality at a time. The x-axis shows how many modalities are given to generate the modality and the y-axis shows the FID of the generated modality.

Figure 21: Plot of finetuning experiment using different $p$ values on how it affects conditional coherence (different score model). The conditional accuracy of the last modality generated by incrementing the given modality at a time. The x-axis shows how many modalities are given to generate the modality and the y-axis shows the accuracy of the generated modality.

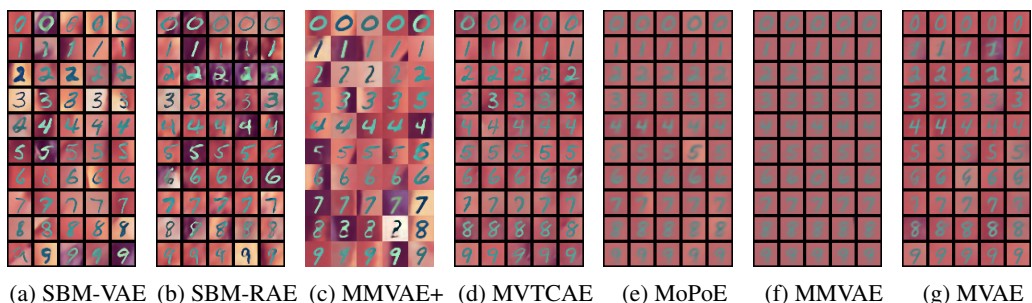

(a) SBM-VAE  (b) SBM-RAE  (c) MMVAE+  (d) MVTCAE  (e) MoPoE  (f) MMVAE  (g) MVAE

Figure 22: Conditional Samples from the third modality given the rest

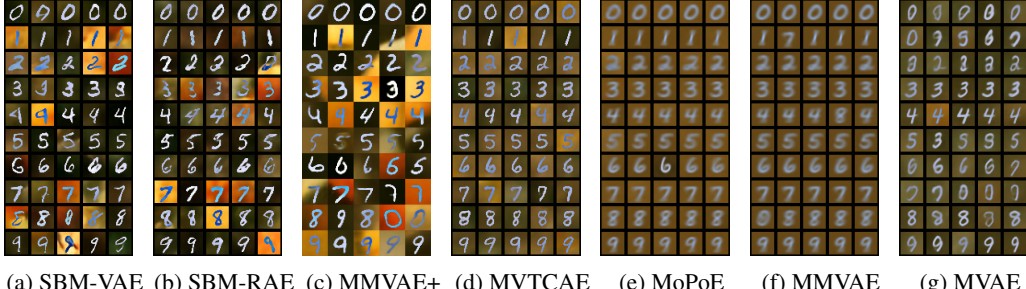

(a) SBM-VAE  (b) SBM-RAE  (c) MMVAE+  (d) MVTCAE  (e) MoPoE  (f) MMVAE  (g) MVAE

Figure 23: Conditional Samples from the first modality given the rest

noises of mean 0 and variance of 0.001, 0.001, 0.1 were added to $z$ before feeding to the decoder for the image, mask, and attribute modality respectively and the best performing one was selected.

The score-based models use a UNET architecture with the latent size reshaped into a size of 16x16. We take the mean of the posterior during training and inference time for SBM-VAE where as the **z** were taken during both times for SBM-RAE. Table 6 shows the detailed hyperparameters used for the score models. DiffuseVAE hyperparameters and models are the same ones used in Pandey et al. (2022) with formulation 1 for the 128x128 CelebHQ dataset. The energy based model uses an MLP network. We train 3 pairs of models for each combination of modality.

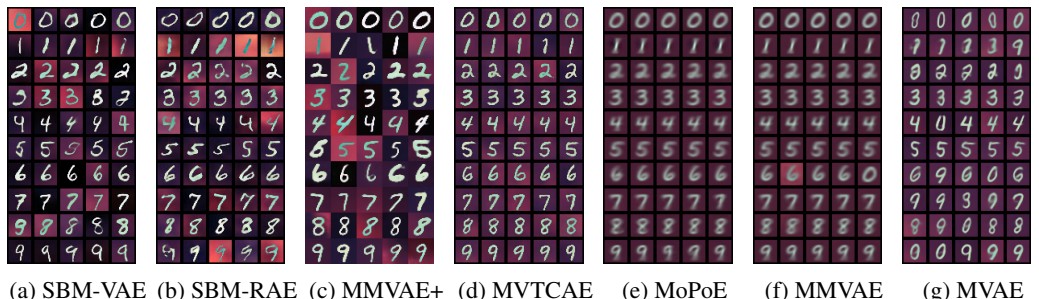

(a) SBM-VAE  (b) SBM-RAE  (c) MMVAE+  (d) MVTCAE  (e) MoPoE  (f) MMVAE  (g) MVAE

Figure 24: Conditional Samples from the sixth modality given the rest

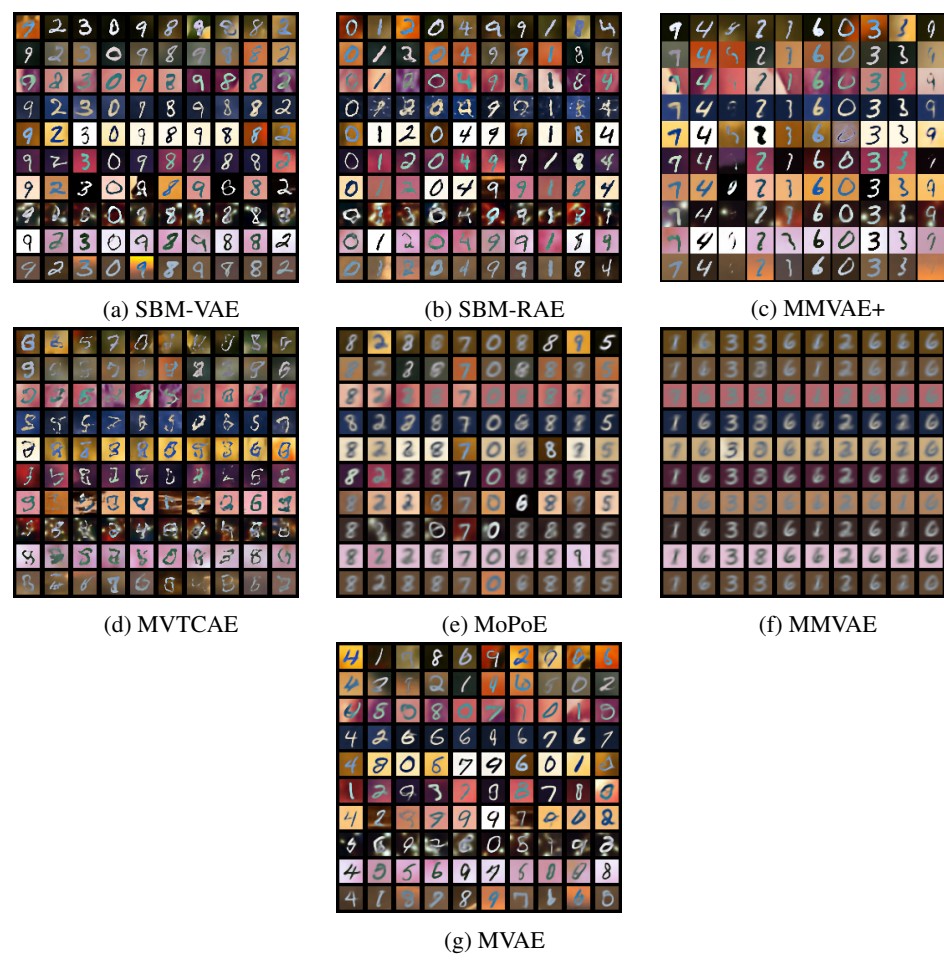

(a) SBM-VAE  (b) SBM-RAE  (c) MMVAE+

(d) MVTCAE  (e) MoPoE  (f) MMVAE

(g) MVAE

Figure 25: Unconditional Samples where each of the columns are unconditional samples from each modality

| Model | $\beta$min | $\beta$max | N | LD per step | EBM-scale | batch-size |
|-------|-------|-------|------|-------------|-----------|------------|
| SBM | 0.1 | 20 | 1000 | 1 | 2000 | 256 |

Table 6: Score Hyperparameters CelebMaskHQ

### A.7 CelebA Ablation

#### A.7.1 High Quality Image generation for CelebAHQ

In our setup, we use a normal variational autoencoder which makes it suitable for baseline comparison and training compute requirements. But since variational autoencoders suffer from low-quality and blurry images, compared to other SOTA generative models such as diffusion models (Dhariwal & Nichol, 2021), in order to get higher-quality, we can do two things. The first is to use a high quality pre-trained auto-encoder and train the score model on it which is possible because we have independent two phase training stages. The second is to further increase the quality of the output of the decoder using DiffuseVAE model (Pandey et al., 2022). We illustrate the latter case here where the generated samples from SBM-VAE are fed into DiffuseVAE as shown in Figure26. The DiffuseVAE helps in generating high quality images from the low quality ones without changing the image characteristics. As the figure shows, the quality of the images is much better while preserving the attributes and the masks given to generate them.

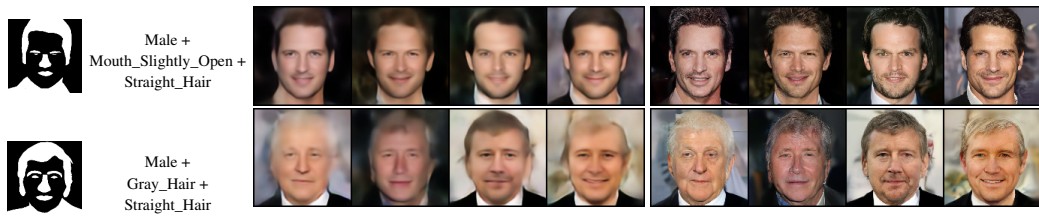

| | | |
|---|---|---|
| Male + Mouth_Slightly_Open + Straight_Hair | | |
| Male + Gray_Hair + Straight_Hair | | |
| | (a) SBM-VAE output | (b) DiffuseVAE output |

Figure 26: Higher quality image generation using DiffuseVAE given mask and attribute shown in the first two columns

#### A.7.2 CelebMaskHQ Qualitative Result

In this section, we show samples from the CelebHQMASK dataset where the generated images are conditioned on different modalities. Figure 27 shows unconditionally generated outputs from each modality. Figure 28 shows different samples where only the image is given, Figure 29 shows different samples where the mask is given, Figure 30 shows different samples where the attribute is given. Figures 32, 31, 33 show different samples where a combination of the two modalities are given and the remaining modality is predicted.

### A.8 Experiment on the MHD Dataset

In this section, we discuss an additional experiment consisting of an audio modality. We use the audio and image modalities from the MHD dataset by Vasco et al. (2022). The image modality is from the MNIST dataset and the audio is a recording of each digit. We use the same encoder-decoder architecture from Vasco et al. (2022) with latent size 64. We conduct experiments on three models: SBMVAE, MVTCAE, and MMVAE+ and we report the result in table 7. The result shows our approach works in the presence of audio modality also.

| Model | Aud -> Img Coherence | Img->Aud Coherence | Unc Coherence |
|---|---|---|---|
| SBMVAE | 0.86 | 0.85 | 0.84 |
| MVTCAE | 0.66 | 0.82 | 0.36 |
| MMVAE+ | 0.45 | 0.44 | 0.25 |

Table 7: MHD Image-Audio result

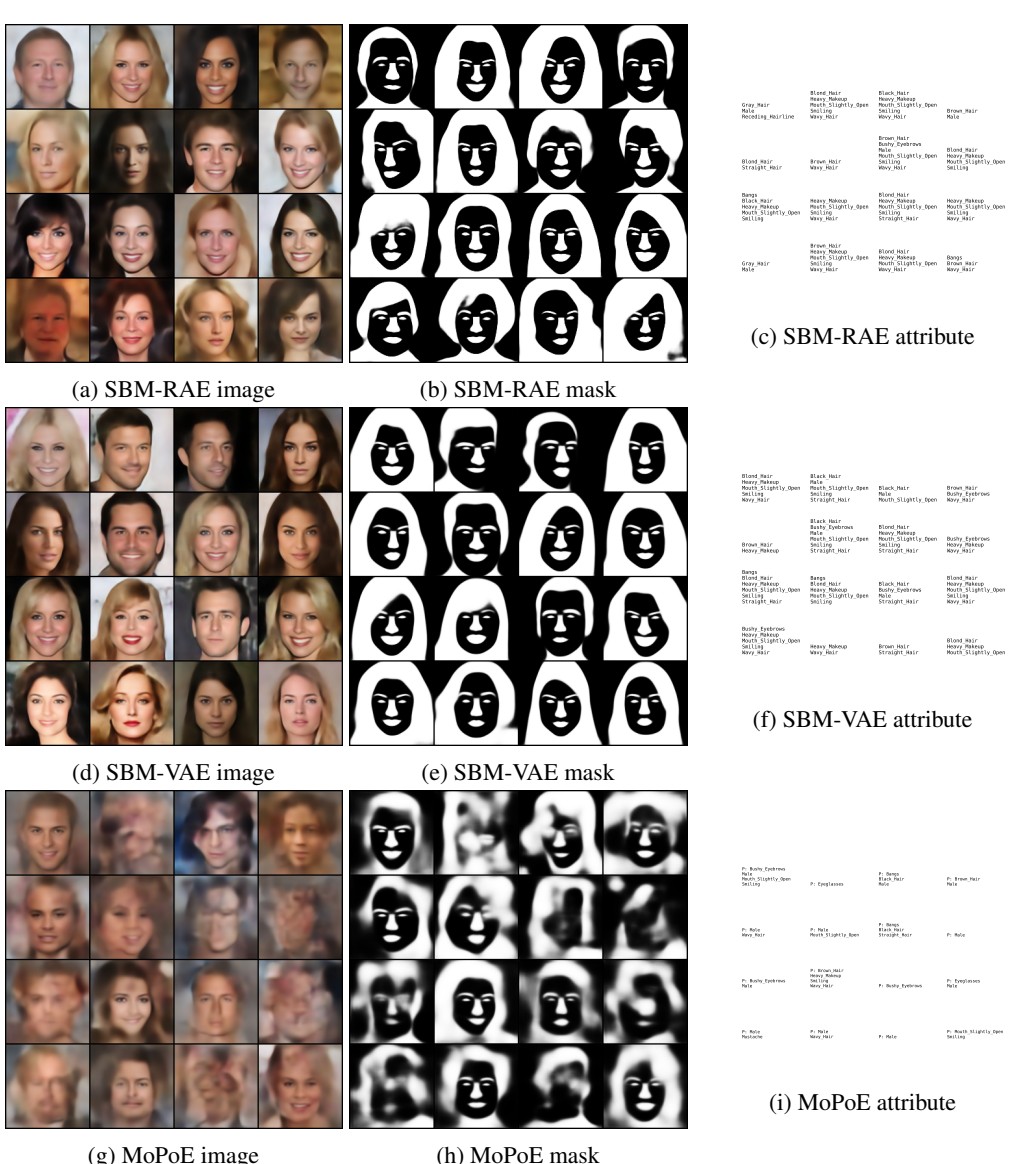

Figure 27: Unconditional generation

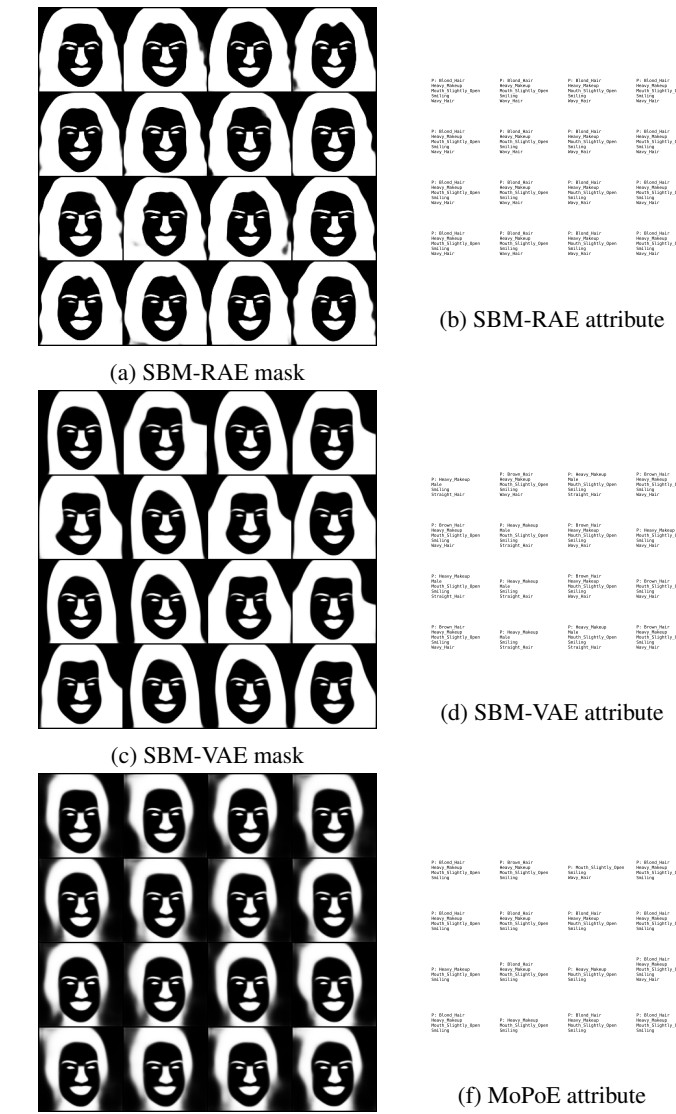

(a) SBM-RAE mask

(b) SBM-RAE attribute

(c) SBM-VAE mask

(d) SBM-VAE attribute

(e) MoPoE mask

(f) MoPoE attribute

Figure 28: Mask and Attribute generation given Image

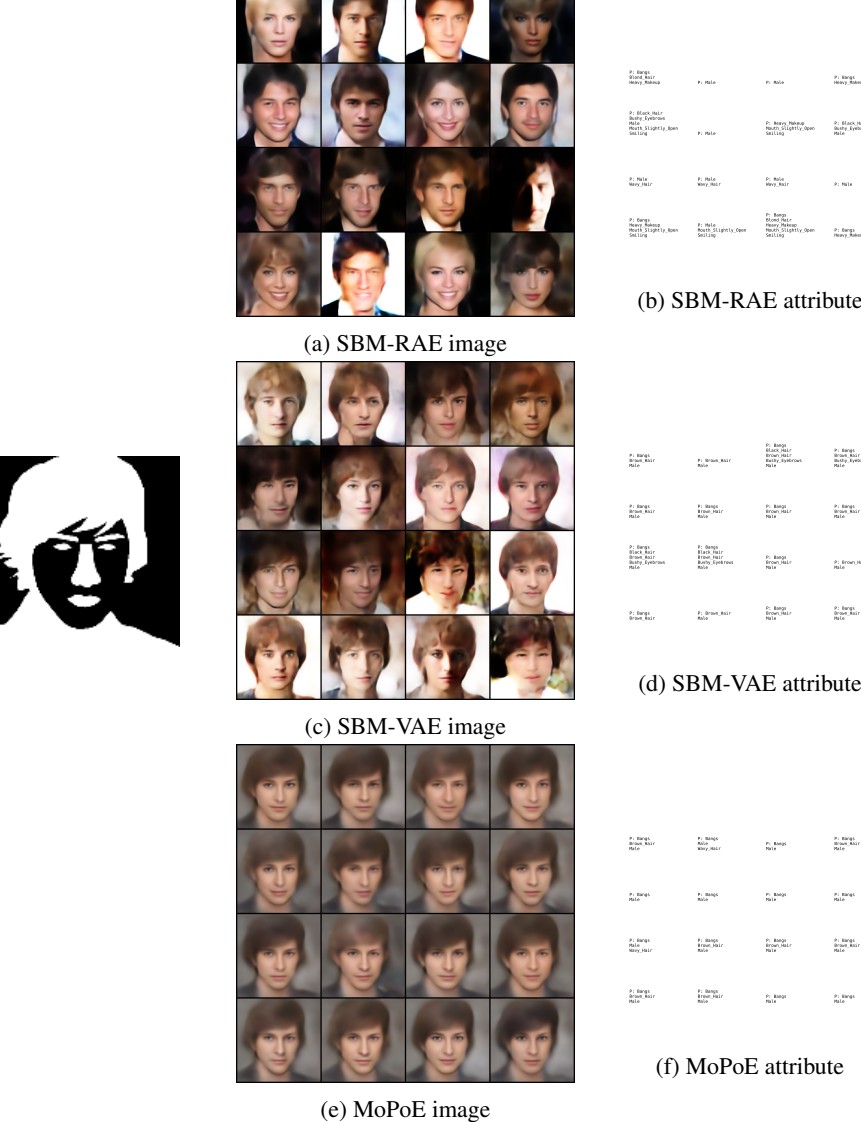

(a) SBM-RAE image

(b) SBM-RAE attribute

(c) SBM-VAE image

(d) SBM-VAE attribute

(e) MoPoE image

(f) MoPoE attribute

Figure 29: Image and Attribute generation given Mask

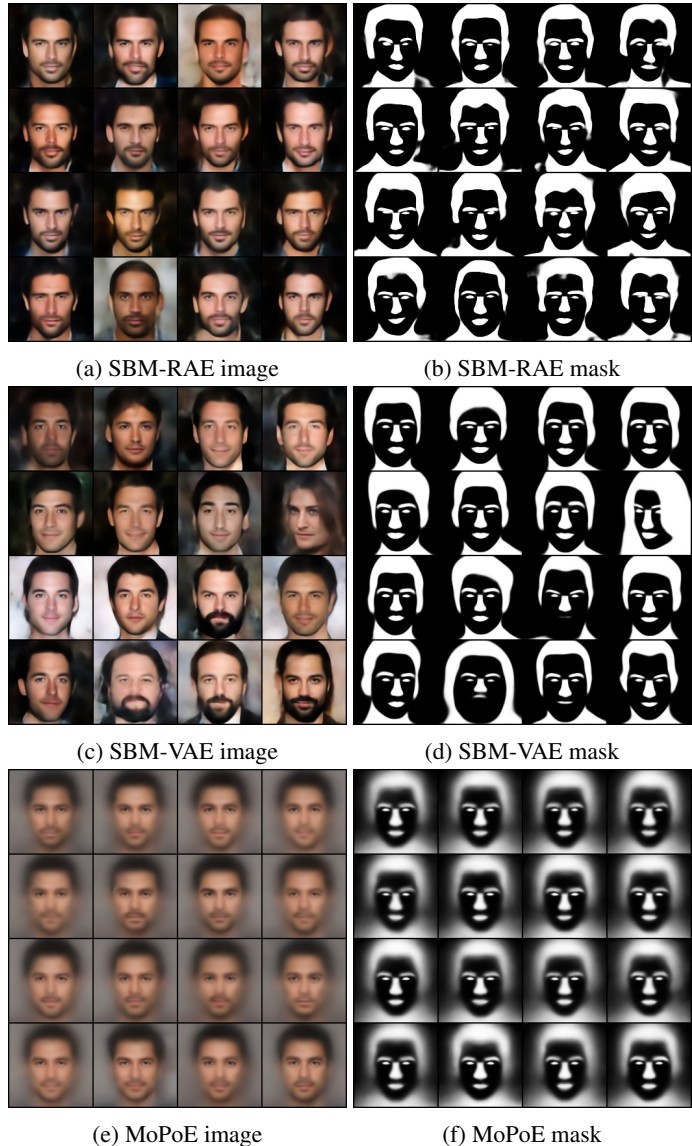

Black_Hair +
Bushy_Eyebrows +
Male +
Mustache +

(a) SBM-RAE image    (b) SBM-RAE mask

(c) SBM-VAE image    (d) SBM-VAE mask

(e) MoPoE image    (f) MoPoE mask

Figure 30: Image and Mask generation given Attribute

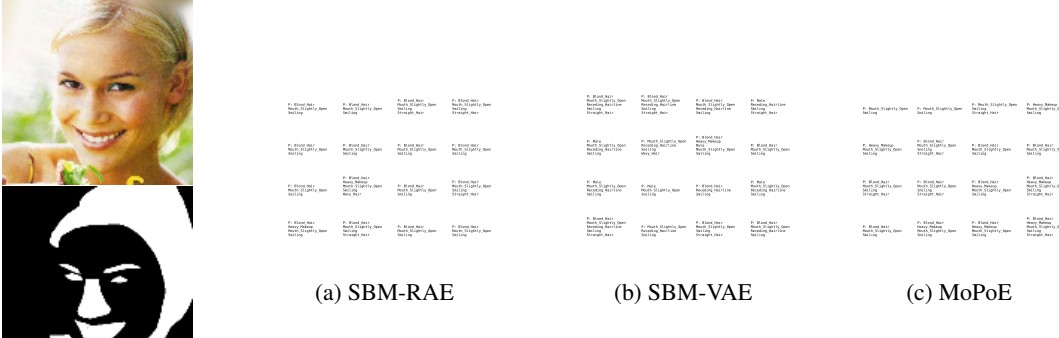

(a) SBM-RAE    (b) SBM-VAE    (c) MoPoE

Figure 31: Attribute generation given Image and Mask

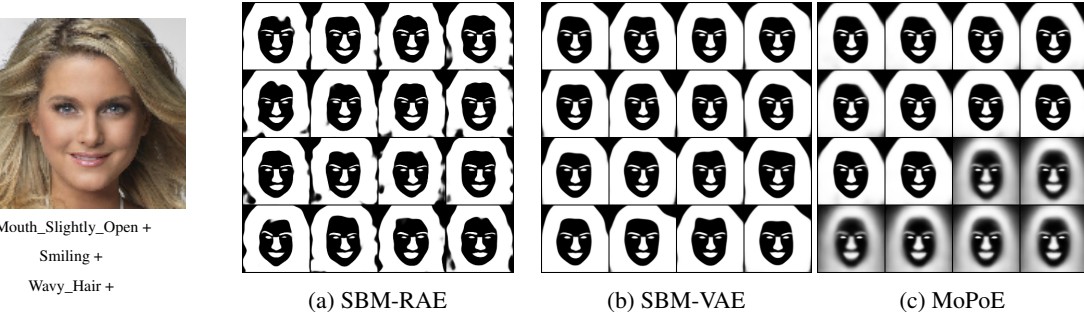

Mouth_Slightly_Open +
Smiling +
Wavy_Hair +

(a) SBM-RAE          (b) SBM-VAE          (c) MoPoE

Figure 32: Mask generation given Image and Attribute

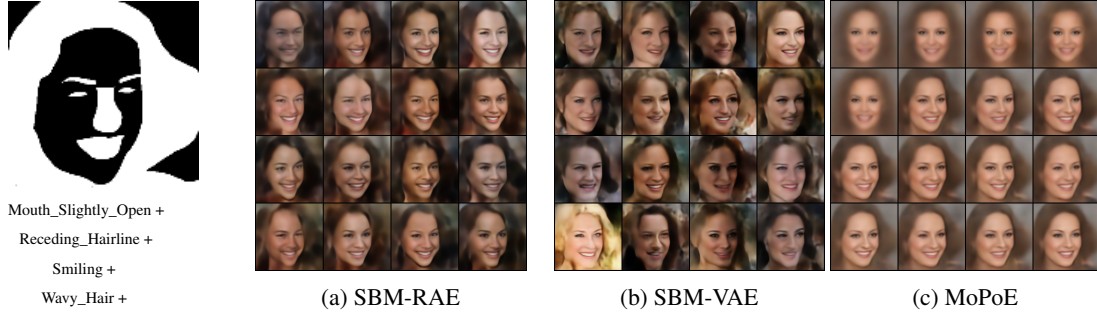

Mouth_Slightly_Open +
Receding_Hairline +
Smiling +
Wavy_Hair +

(a) SBM-RAE          (b) SBM-VAE          (c) MoPoE

Figure 33: Image generation given Mask and Attribute

