# OpenReview forum: "Score-Based Multimodal Autoencoders"
_ICLR.cc/2024/Conference — Submitted to ICLR 2024_

### Official Review · Reviewer_L126 · 2023-10-31

**Soundness:** 2 fair
**Presentation:** 2 fair
**Contribution:** 2 fair
**Rating:** 3
**Confidence:** 3

**Summary:**

The authors propose a new multimodal VAE model consisting of unimodal VAEs and a score-based model that models the joint distribution of unimodal latent variables. They also introduce an energy-based coherence guidance model, which helps to alleviate the problem when the predicted modalities are not aligned with the observed modalities. Experiments are performed on PolyMNIST and CelebAMask datasets.

**Strengths:**

- Clearly explained expected properties of a multimodal generative model.
- Proposed method scales with the number of modalities which is important for multimodal generation.
- Extensive related work.

**Weaknesses:**

- Missing connection and comparison of the methods in the related work section to the proposed method. In what way are SBM-* improving over these methods? I also find that such discussion is missing in the experimental results.
- No ablation studies on the components of the proposed method. This would also help gaining a more intuitive understanding of the proposed method, which is lacking in the current version.
- Limited experimental evaluation using only image data. Both PolyMNIST and CelebAMask are image datasets. It would be beneficial to add experiments on text audio modalities as well.
- I find the novelty of the method a bit limiting. Given that the experimental results do not report any standard deviations makes it hard to judge the weight of the contribution.

**Questions:**

- It has been shown by several works that FID does not provide an adequate evaluation of a generative model (see for example work by Sajjadi et al Assessing Generative Models via Precision and Recall). Are you sure that none of the considered models experience a mode collapse? It would be better to present these results with newer metrics like Improved Precision and Recall [1] or Delaunay Component Analysis [2] below). With the latter, you could also gain more insights into each generated modality by analysing the geometry of their representations.

[1] Kynkäänniemi et al, Improved Precision and Recall Metric for Assessing Generative Models, NeurIPS 2019

[2] Poklukar et al, Delaunay Component Analysis for Evaluation of Data Representations, ICLR 2022

- When calculating conditional accuracy, do you randomise the observed modalities? For example, in Fig 5, how many times did you repeat this experiment? I believe that without std it is hard to draw conclusions.

- I do not see the benefits of keeping both SBM-VAE and SBM-RAE in the experiments. First, it is not clear what version is better and why. The authors do not discuss this at all. Second, it hinders the readability of the results.

- I do not fully understand the unconditional coherence evaluation. Do you sample z_1:M, then generate all x_1:M modalities and evaluate their agreement ? In Fig 6 what are ”similar modalities”? Please add some more explanation.

---

> ### Author Response · Authors · 2023-11-21
> **Rebuttal by Authors**
>
> Dear reviewer,
>
> Thank you for your comments.
>
> *“Missing connection and comparison of the methods in the related work section to the proposed method. In what way are SBM improving over these methods?...“*
>
> Multimodal VAEs (MoPoE, MVAE, MMVAE, MMVAE+, MVTCAE) learn a joint representation (newer approaches like MMVAE+ added modality-specific representation as well) and forcing the decoder to use this joint representation reduces the generation quality of each modality while helping to improve the coherence (we have seen a similar effect with fine-tuning our decoder to increase coherence see Section A.4.2). Our work, on the other hand, does not learn a joint representation but learns a joint distribution over individual representation, this way the decoder and individual learned representation remains intact and decoder uses only modality-specific representation. In addition, our method exhibits superior coherence in unconditional generation. When no modality has been observed, previous work fails to recover the joint representation. In contrast, our approach samples from a valid joint distribution, resulting in enhanced coherence.
>
> *“No ablation studies on the components of the proposed method.”*
>
> The main components of our approach are the score-based model and coherence-guidance EBM. We already have included the results with and without EBM and we have studied the scalability of the score-based model in the presence of a different number of modalities (when trained with 2, 3, all the way up to 10 modalities shown in Figure 8 - (a,b,c)).
> In the revised version, we also added Table 4 to the paper to explain the effects of beta on the score-based model.
> Additionally, we added another section to explore the effect of decoder fine-tuning to improve the coherence.
> If you can think of other ablation studies that are missing from our paper, we are willing to include them.
>
> *“Limited experimental evaluation using only image data. Both PolyMNIST and CelebAMask are image datasets. It would be beneficial to add experiments on text audio modalities as well”*
>
> We chose PolyMnist for its simplicity in performing experiments on multiple modalities. Studying the behavior of multimodal VAEs in the presence of different numbers of missing modalities requires a tractable dataset and problem setting.
> We also added CelebA-MASK-HQ with the attributes as an additional non-image modality. In addition to that, we have added audio-image modality from the MHD dataset [1] (please see the revised version).
>
> *“I find the novelty of the method a bit limiting. Given that the experimental results do not report any standard deviations makes it hard to judge the weight of the contribution.”*
>
> In our graphs, we plot the standard deviation as a shade in the figures (more visible when zoomed). We report the mean over three-experiment runs on our figures and tables. We have edited the table to include the standard deviation.
>
>
> *“It has been shown by several works that FID does not provide an adequate evaluation of a generative model…”*
>
> Despite the extensive research for introducing alternative metrics for evaluating generative quality for images, FID has been extensively used in different generative model evaluations [2,3,4] (to name a few), including in all of the baselines in Multimodal VAEs, mostly to achieve comparison consistency.
>
> *“It would be better to present these results with newer metrics like Improved Precision and Recall [1] or Delaunay Component Analysis [2] below)...”*
>
> Thank you for your suggestions, however, it is worth noting that our method is not learning a joint representation for data modalities and we rely on the learned representation from individual VAEs for each modality. We are claiming that learning the correlation among latent representations of individual modalities is enough to achieve coherence.
>
> *“ Are you sure that none of the considered models experience a mode collapse? “*
>
> We have added section A.5 to investigate the possibility of mode collapse. We unconditionally generate all the modalities 10000 times and count the present labels using a pretrained classifier. All methods cover all modes (different digits) for each modality, however, MVTCAE has less balanced coverage compared to our approach and MMVAE+ .
>
> *“When calculating conditional accuracy, do you randomize the observed modalities? For example, in Fig 5, how many times did you repeat this experiment? I believe that without std it is hard to draw conclusions.”*
>
> We have reported the standard deviation as shades under the main graph (we ran each  3 times). We use the whole test dataset to calculate the results shown, so yes, multiple data points are given to generate the results. We also revised the tables to include standard deviation.

---

> > ### Author Response · Authors · 2023-11-21
> > **Continued ...**
> >
> > *“I do not see the benefits of keeping both SBM-VAE and SBM-RAE in the experiments.”*
> >
> > Actually, there is no dominant version across all experiments. Our method is independent of how the individual latent representations are being learned, so we show this flexibility by combining it with both variational autoencoders and regularized deterministic autoencoders.
> >
> > *“I do not fully understand the unconditional coherence evaluation”*
> >
> > Yes, your interpretation is correct. They are first initialized from Gaussian noise. We then sample from the score-based models by starting from the noise and iteratively updating the samples. Final samples are fed to each decoder to generate images. The similar modalities are the number of modalities that agree (same digit) as classified by a pre-trained classifier.
> >
> > [1] M. Vasco, H. Yin, F. S. Melo, and A. Paiva. Leveraging hierarchy in multimodal generative models for effective cross-modality inference. Neural Networks, 146:238–255, 2 2022.
> > [2] Rombach, R., Blattmann, A., Lorenz, D., Esser, P., & Ommer, B. (2021). High-Resolution Image Synthesis with Latent Diffusion Models. CoRR, abs/2112.10752. Retrieved from https://arxiv.org/abs/2112.10752
> > [3] Song, Y., Sohl-Dickstein, J., Kingma, D. P., Kumar, A., Ermon, S., & Poole, B. (2021). Score-Based Generative Modeling through Stochastic Differential Equations. arXiv [Cs.LG]. Retrieved from http://arxiv.org/abs/2011.13456
> > [4] Imant Daunhawer, Thomas M. Sutter, Kieran Chin-Cheong, Emanuele Palumbo, and Julia E Vogt. On the limitations of multimodal VAEs. In International Conference on Learning Representations, 2022. URL https://openreview.net/forum?id=w-CPUXXrAj.

---

### Official Review · Reviewer_7BDS · 2023-10-31

**Soundness:** 3 good
**Presentation:** 2 fair
**Contribution:** 2 fair
**Rating:** 6
**Confidence:** 4

**Summary:**

This paper proposes a novel multimodal VAE model which addresses the problem of sample quality deterioration as the number of modalities increases, which the previous models are suffering from. The paper proposed to model separate encoder-decoder pipelines and independent posterior distributions for latent variable corresponding to separate modality to improve the generation quality. The author used a Score-based model (SBM) approach to model a joint prior distribution of latent variables for different modalities to achieve coherence among the generated modalities. The authors then performed benchmark on the modified PolyMnist and CelebAMask-HQ dataset.

**Strengths:**

The author provides a good summary of existing work in the introduction and provided a clear explanation of the motivation of their approach.

The author was able to provide evidence of improved generative quality in terms of FID scores and qualitative analysis on the two datasets.

The paper provides interesting, and sound use case of SBM to learn a joint latent prior distribution for inference and provide cases for conditional inference with sets of observed modality and unconditional inference when no modality is observed, along with an EBM based guidance to enhance coherence for sampling.

**Weaknesses:**

Concerns on the benchmark datasets are too simple, because these datasets have very fixed pattern characteristics, and the modality types are also relatively fixed. Therefore, it might not be sufficient to prove the stability and quality of the proposed method in the case of multi-modal and missing modes.

Details about the specific network structure in the appendix are not clear and dimensions of the latent variables are not clearly stated also.

The evaluation metric for accuracy is not clearly defined.

It is not clear if the generative quality is truly decreasing as the number of modalities increases. This relation is not clear in other models (as the generative quality though less optimal than the SBM approach, their generative quality is largely consistent along the number of modalities).

The coherence of the SBM based model does not show a clear improvement or sometimes performed not as well as previous model (especially in experiment on CelebA-Mask-HQ dataset) in terms of coherence.

The author used a modified PolyMnist dataset but not clearly defined how the new modalities are generated.

**Questions:**

Please define the accuracy metric. Or is it a term used interchangeably with metric coherence?

Please specify the dimensions of the latent variables, are those 1D vectors or 2D matrices? How is Unet applied in this case?

The reviewer acknowledges that the generative quality in terms of FID and qualitative analysis shows an improvement, however the coherence is not improving compared to other works. It would be great if the authors could justify why better generative quality is worth sacrificing for less optimal performance for coherence in the context of multimodality learning.

It would be great if the author could provide more on how the modified PolyMnist dataset is generated consider this is a new version of the original dataset.

---

> ### Author Response · Authors · 2023-11-21
> **Rebuttal by Authors**
>
> Dear Reviewer,
>
> Thank you for your comments.
>
> *“Concerns on the benchmark datasets are too simple, because these datasets have very fixed pattern characteristics, and the modality types are also relatively fixed.”*
>
> Extended PolyMnist enables us to study our approach and baselines in the presence of different numbers of missing modalities. The number of required experiments for such a study is prohibitive for using a more complex benchmark.
> To showcase the performance of our approach on a high-dimensional setting we have used the high-dimensional CelebA-Mask-HQ dataset which is far more difficult and also includes attribute modality.
> In addition, to address the reviewer's concern regarding the type of modalities, we have added audio-image modality from the MHD dataset by [5] (please see Section A.8)
>
> *“Details about the specific network structure in the appendix are not clear and dimensions of the latent variables are not clearly stated also.”*
>
> We have stated the dimensions of the latent variables in Appendix A.2. We have used a latent size of 64 for our models and MMVAE+ and we increased the latent dimension of the other baselines to 64 ∗ n where n is the number of modalities.
> For example, for 10 modalities, we used a latent size of 64 for unimodal VAEs of SBMs and MMVAE+ (because it uses modality-specific latent representation), but for the other baselines, we used a latent size of 640 dimensions to have a fair comparison.
>
> *“The evaluation metric for accuracy is not clearly defined.”*
>
> The coherence is modality-specific. For extended PolyMNIST, when there is no observed modality we have reported the number of agreeing modalities as coherence. When there is an observed modality, the coherence reduces to the agreement of the predicted modality with the observed modality, which is accuracy. For example, for each image in the observed modality, we checked if the label of the predicted target modality agrees with the label of the observed modality, and reported the test-set average. To obtain the label of predicted modality we have used a pre-train classifier.
> For CelebA-Mask-HQ, we used the test set average of the F1 score between the predicted attributes (or mask pixel) and true attributes (or mask pixel) in the test set given the observed modality (e.g. image).
>
>
> *“It is not clear if the generative quality is truly decreasing as the number of modalities increases.”*
>
> Generative quality decreases (FID increases) slightly for Product of Expert (PoE) models which use additional information as the number of observed modalities increase but remains fairly flat for mixture models which sample modalities from observed ones.
>
> *“The coherence of the SBM based model does not show a clear improvement or sometimes performed not as well as previous model (especially in experiment on CelebA-Mask-HQ dataset) in terms of coherence.”*
>
> The main reason is that the SBM-AE learns to generate samples from the missing modalities and for structured prediction tasks such as mask prediction or attribute prediction, the samples may be different slightly, therefore our approach has been penalized by the used F1 score metrics. On the other hand, using joint representation results in average outputs which works in the favor of the other multi-modal VAEs.
>
> *“The reviewer acknowledges that the generative quality in terms of FID and qualitative analysis shows an improvement….”*
>
> Yes, that’s correct, but as [4] also states, the generation quality of multimodal VAEs makes them unusable in real-world applications. So, the main contribution was on the quality while having comparable coherence to previous methods. Our models also outperform previous works in terms of unconditional coherence and quality.
>
> *“It would be great if the author could provide more on how the modified PolyMnist dataset is generated consider this is a new version of the original dataset.”*
>
> The new modalities are generated the same way as PolyMnist by changing the background of the Mnist dataset as stated in Appendix A.2. We have attached the backgrounds used, the dataset, and the code that was used to generate them in the zip file.
>
>
> *“How is Unet applied in this case?”*
>
>  Unet was used by resizing 64 dimensions of PolyMnist to 8x8 and 256 dimensions of CelebA-Mask-HQ to 16x16 as mentioned in Appendix A.2 and A.5 respectively.
>
>
> [1]Thomas M. Sutter, Imant Daunhawer, and Julia E. Vogt. Generalized multimodal ELBO. In 9th International Conference on Learning Representations, ICLR 2021, Virtual Event, Austria, May 3-7, 2021. OpenReview.net, 2021. URL https://openreview.net/forum?id= 5Y21V0RDBV

---

> > ### Author Response · Authors · 2023-11-21
> > **Continued ...**
> >
> > [2]Yuge Shi, Siddharth Narayanaswamy, Brooks Paige, and Philip H. S. Torr. Variational mixtureof-experts autoencoders for multi-modal deep generative models. In Hanna M. Wallach, Hugo Larochelle, Alina Beygelzimer, Florence d’Alché-Buc, Emily B. Fox, and Roman Garnett (eds.), Advances in Neural Information Processing Systems 32: Annual Conference on Neural Information Processing Systems 2019, NeurIPS 2019, December 8-14, 2019, Vancouver, BC, Canada, pp. 15692–15703, 2019. URL https://proceedings.neurips.cc/paper/2019/hash/ 0ae775a8cb3b499ad1fca944e6f5c836-Abstract.html.
> >
> > [3]Thomas M. Sutter, Imant Daunhawer, and Julia E. Vogt. Multimodal generative learning utilizing jensen-shannon-divergence. In Hugo Larochelle, Marc’Aurelio Ranzato, Raia Hadsell, MariaFlorina Balcan, and Hsuan-Tien Lin (eds.), Advances in Neural Information Processing Systems 33: Annual Conference on Neural Information Processing Systems 2020, NeurIPS 2020, December 6-12, 2020, virtual, 2020. URL https://proceedings.neurips.cc/paper/2020/ hash/43bb733c1b62a5e374c63cb22fa457b4-Abstract.html.
> >
> > [4] Imant Daunhawer, Thomas M. Sutter, Kieran Chin-Cheong, Emanuele Palumbo, and Julia E Vogt. On the limitations of multimodal VAEs. In International Conference on Learning Representations, 2022. URL https://openreview.net/forum?id=w-CPUXXrAj.
> >
> > [5] M. Vasco, H. Yin, F. S. Melo, and A. Paiva. Leveraging hierarchy in multimodal generative models for effective cross-modality inference. Neural Networks, 146:238–255, 2 2022.

---

> > ### Comment · Reviewer_7BDS · 2023-11-23
> >
> > Dear Authors,
> >
> > Thank you for the response. The reviewer appreciate the your response to our concerns and found the clarifications was able to address the confusion and the confusion of the model architecture, metrics and dimensions of latent variables. However, there are still a few issues remain.
> >
> > 1. For the reviewer's original concern on "It is not clear if the generative quality is truly decreasing as the number of modalities increases. ..." The reviewer understand the author pointed out certain model shows such trend, but consider the subtly of the increase universally across the models and the limited testing on just two datasets it would not be rigorous enough to make such claim. But the review do acknowledge the SBM-XAE models enhances the generative quality and thank you for the reference one the importance of the quality factor.
> >
> > 2. It would be great if the author could further elaborate on why F1 metric is not in favor of the SBM-XAE. Also, consider the use case of  the conditional denoising SBM, why does the author have to use separate VAEs for separate modalities. Why not add a VAE that jointly encode and decode the modalities? Following the author's rebuttal argument, is that going to further enhance the coherence performance?
> >
> > 3. The reviewer appreciate the new experiment on the MHD dataset. We notice there is a big increase in the relative performance for the coherence metric relative to other models. Could the author elaborate why the model suddenly performs much better than other SOTA models?
> >
> > The reviewer again thank the author for their response to our concerns.

---

> > > ### Author Response · Authors · 2023-11-23
> > >
> > > Dear Reviewer,
> > >
> > > We are glad that we could address some of your concerns and thank you for pointing out the issues that require further clarification.
> > >
> > > 1. We should point out that we have two types of modality increment, one is having a different number of modalities at inference time in which the model has been trained on the fixed number of modalities and queried based on observing different numbers of modalities. The other is training on a different number of modalities, for which the results have been reported in Figure 8. We acknowledge that our experiments are not comprehensive enough to conclude that inference time generative quality of multimodal VAEs dropped consistently with the number of modalities, and we will relax that statement. However, the train time drop in the generative quality as we increase the number of modalities has been theoretically and experimentally supported by Daunhawer (2022) and is evident in Figure 8. It is also worth mentioning that we have increased the latent capacity of baseline multimodal VAEs that do have modality-specific representation, proportional to the number of modalities (e.g. the latent size of MoPoE is n*64, where n is the number of trained modalities). This technique has reduced the decline in their generative quality.
> > >
> > > 2. Assessing the F1 score for attribute prediction or mask prediction essentially gauges the conditional coherence between the predicted modality and the provided image, rather than evaluating the generative quality. This approach disadvantages our methods that do not leverage a joint representation, especially since there exists only one correct mask or attribute set for each image. Our method, while not yet matching the conditional coherence of SOTA multimodal VAEs, introduces a novel direction that prioritizes high-quality generation, with the potential for further improvements to bridge this gap. As suggested, a direction for enhancement could involve independently training a joint representation and utilizing it to condition the SBM.
> > >
> > > 3. The behavior of our method on MHD is consistent with PolyMNIST experiments. With only 2 modalities present, our method shows better coherence than the baselines as shown in Figure 8.

---

### Official Review · Reviewer_NJdA · 2023-11-01

**Soundness:** 2 fair
**Presentation:** 2 fair
**Contribution:** 3 good
**Rating:** 6
**Confidence:** 5

**Summary:**

To improve over existing multimodal VAEs, the paper proposes to train unimodal VAEs independently, in order to achieve high generative quality, and a score-based model to learn a joint latent space across modalities and achieve semantic coherence.

**Strengths:**

- The paper deals with the relevant problem of tackling the current limitations of multimodal VAEs.
- Interesting and encouraging results are shown to back up the effectiveness of the proposed approach. In particular, it is encouraging that the performance of the proposed approach can benefit from the presence of more modalities in terms of performance, and does not suffer from the limitations uncovered in recent work [1].
- The comparisons reflect the state-of-the-art in the field, with recent approaches included.


[1] Daunhawer I, Sutter TM, Chin-Cheong K, Palumbo E, and Vogt JE. On the limitations of multimodal VAEs. In International Conference on Learning Representations, 2022.

**Weaknesses:**

- To me it is unclear why the score-matching model would lead to a coherent shared latent space, and I would appreciate if authors would clarify that. While in the first step of training we have a prior $p(z_{1:M})=\prod \mathcal{N}(0, \sigma I)$, what are the modelling assumptions on the parametric prior $p_{\theta}(z_{1:M})$ in the second step?
- The comparison with MVTCAE [2] from e.g. Figures 4 and 5 is quite important and should be further commented, with maybe more insights (eg. different $\beta$ values, average performance across modalities for conditional generation, see below) for at least two reasons. First as it does not sub-sample modalities during training, MVTCAE is not subject to the limitations for generative quality uncovered in [1]. Second, MVTCAE and the proposed model are really similar in performance, and from the results it is unclear which one performs better overall (at least in the first experimental setting).
- Why reporting only the results on the last modality in e.g. Figure 4? (Even though also results on the first modality are available, in the Appendix).This is only a partial insight on model performance, and results should be averaged across modalities for conditional generation (not modalities used for inference, modalities used for generation) to show performance is consistently good. To back up this point, results on the third modality (Figure 3) indicate that MVTCAE has poor generative quality. However, FIDs in Figure 4 for the last modality indicate otherwise. Hence, it seems that in evaluating model performance, one might want to control for the effect of choice of modality.
- Did the authors do an ablation for different values of $\beta$ for the compared models in e.g. Extended PolyMNIST? In the Appendix it is stated that the $\beta$ was chosen using the validation set. How exactly was it chosen? Should it be by looking at the ELBO value on the validation set, one should be careful as likelihood values prove often not to be representative when it comes to performance of multimodal VAEs. For instance, with models that subsample modalities, high likelihoods for conditional reconstruction across modalities can be obtained by producing average-looking samples (since the modality-specific information about the sample to be reconstructed cannot be inferred). I think it would be important to report results for the compared models across $\beta$ values instead of reporting results only for a single value. To back up this point, the conditional generation results from Figure 3 for e.g. MVTCAE seem rather different from what reported original work [2] on PolyMNIST, in which the authors use a much higher $\beta$ value ($\beta=2.5$ I think).
- I found many typos and imprecisions in the manuscript, that has margin for improvement in writing quality.

[1] Daunhawer I, Sutter TM, Chin-Cheong K, Palumbo E, and Vogt JE. On the limitations of multimodal VAEs. In International Conference on Learning Representations, 2022. [2] Hwang HJ, Kim GH, Hong S, and Kim KE. Multi-view representation learning via total correlation objective. In Advances in Neural Information Processing Systems, 2021.

**Questions:**

- I would change "alternative" to "novel" in the Abstract. Not really clear what the approach is "alternative" to otherwise.
- In the Introduction, I can suggest to make the difference between "Scalability" and "Conditional modality gain" clearer. As I understand it, scalability refers to the fact that a multimodal VAE trained on a given number of modalities should perform at least as well as the same model trained on fewer modalities. On the other hand, conditional modality gain refers to test-time conditional generation performance improving when more modalities are given for inference.  I would suggest to make a clearer distinction in the text between the two concepts.

For questions and suggestions to the authors, see also "Weaknesses" section.

---

> ### Author Response · Authors · 2023-11-21
> **Rebuttal by Authors**
>
> Dear Reviewer,
>
> Thank you for your comments.
>
> *“To me it is unclear why the score-matching model would lead to a coherent shared latent space, and I would appreciate if authors would clarify that…”*
>
> Our approach leverages the fact that SBM implicitly learns the underlying data distribution by capturing the correlation among their input variables, which are here the latent representation learned by uni-variate VAEs. SBMs offer tractable joint or marginal sampling, which has been used to sample the latent representation of a missing modality. Therefore, our approach does not require learning an explicit joint representation which is common in the prior multi-modal VAEs.
>
>  *“what are the modelling assumptions on the parametric prior in the second step?”*
>
> In the second step we learn the parametric prior which has been parameterized by a UNET.
> We don’t enforce any specific assumption for training the SBM.
>
> *“dfferent $\beta$ values, average performance across modalities for conditional generation...”*
>
> We have added Table 3 in the paper listing the average performance (average of each modality given the rest) of each model as well as the performance for individual modalities in Figure 13 and Figure 14.  MVTCAE results in better accuracy while having a higher FID score due to existing joint representation. When none of the modalities has been observed (unconditional generation), our approach is superior to all of the prior multimodal VAEs including MVTCAE. This is a very important property in scenarios like generating a piece of coherent music consisting of multiple instruments (with each instrument as a modality).
>
> *“Why reporting only the results on the last modality in e.g. Figure 4?” *
>
> We have added Table 3, Figure 13, and Figure 14 in the revised version.
>
> *“Did the authors do an ablation for different values of beta for the compared models in e.g. Extended PolyMNIST”*
>
> Please see the revised section of A.4.2
>
> *“In the Appendix it is stated that the model was chosen using the validation set. How exactly was it chosen”*
>
> Models were selected using a combination of the task metrics (fid and coherence in both conditional and unconditional settings) to address the existing tradeoff. Note that MVTCAE has a tradeoff between conditional and unconditional performance with respect to different beta. For example, MVTCAE performs better conditionally with lower beta but worse unconditionally. The experiments can be replicated using the zip file we have attached.
>
> *“I would change "alternative" to "novel" in the Abstract. Not really clear what the approach is "alternative" to otherwise.”*
>
> We employed the term 'alternative' to emphasize the distinctiveness of our method compared to conventional Multimodal VAE techniques, which typically rely on joint representations.
>
> *"Scalability" and "Conditional modality gain":*
>
> Thank you for pointing that out. We refer to Scalability as computational efficiency when adding modalities while performing comparably to a model trained with a lower number of modalities. We have edited that line to make it more clear.

---

### Official Review · Reviewer_qLtQ · 2023-11-06

**Soundness:** 3 good
**Presentation:** 4 excellent
**Contribution:** 3 good
**Rating:** 8
**Confidence:** 3

**Summary:**

In this study, the researchers introduce a multimodal modeling framework that utilizes distinct encoder-decoder pairs for each type of data input. They devise a method to learn a shared prior for the latent variables that represent different modalities, in order to understand the interconnections between them. Each modality's training is optimized through the maximization of its Evidence Lower Bound (ELBO). The latent variables for each modality are then independently generated using the unimodal encoders. These variables are jointly modeled using a denoising score-matching technique. Additionally, the latent variables are trained to be consistent with each other through a noise-contrastive approach, and the gradient of the noise-contrastive model is used to adjust the scores for latent variables that are not observed.

Using their proposed method, the authors demonstrate that the Frechet Inception Distance (FID) score of the created outputs remains stable, even as the number of modalities grows. Moreover, the congruence between the modalities, measured by how accurately the class label of the anticipated modality can be predicted, is enhanced when the model includes a mechanism for coherence guidance.

**Strengths:**

1) The paper is well-written and easy to read.

2) The primary novelty of the paper stems from its simplicity. While the previous papers have been focussed on modeling all the modalities together, this paper decomposes the problem into two independent aspects
 - Learn latent variables for each modality individually
 - Learn a joint distribution over the latent variables.
This is reminiscent of Dall-E [1] where a separate discrete VAE is learned for images followed by a mapping from the text to the latent variables. In [2], a separate VAE is learned for each modality while simultaneously minimizing the KL divergence between the latent variables. However, none of these approaches can be extended directly to more than 2 modalities.

This paper is most reminiscent of the MMVAE+ paper that learns unimodal as well as cross-modal features.

3) The results in this paper (particularly, the idea of learning latent variables independently) can be significant for the multimodal community.


[1] Ramesh, Aditya, et al. "Zero-shot text-to-image generation." International Conference on Machine Learning. PMLR, 2021.
[2] Pandey, Gaurav, and Ambedkar Dukkipati. "Variational methods for conditional multimodal deep learning." 2017 international joint conference on neural networks (IJCNN). IEEE, 2017.

**Weaknesses:**

1) The proposed approach assumes that all the modalities are present during training. It can't be used for training with missing modalities, unlike other joint learning-based approaches. However, in my opinion, this is not  a major concern
2) Since the latent variables for each modality are learned independently, they can be highly misaligned. Perhaps, the authors must consider incorporating information from coherence-guided EBM while training the latent variables.
3) Equation (5) doesn't make much sense since z_u is present in the LHS while also getting marginalized in RHS.

**Questions:**

None

---

> ### Author Response · Authors · 2023-11-21
> **Rebuttal by Authors**
>
> Dear Reviewer,
>
> Thank you for your comments.
>
> *“The proposed approach assumes that all the modalities are present during training…”*
>
> Yes, as highlighted in our submission, our current model operates under the assumption that all modalities are present. For future work, we plan to adapt our model to train effectively even when certain modalities are missing. This may be accomplished by modifying the score matching objective to mask missing modalities or by imputing the missing modality using sampling from the so-far trained model.
>
>
> *“Since the latent variables for each modality are learned independently, they can be highly misaligned…”*
>
> We acknowledge that the latent representation learned by our model may not be aligned. Indeed, incorporating an alignment constraint could potentially simplify the training process for the score model, and we plan to explore this possibility in future research. Nonetheless, a strong and robust score model captures the correlation among the latent representations of various modalities. This is particularly important for our SBM-AE framework, as it is designed to work with any pre-trained VAE. Moreover, relying on coherence guidance from an EBM might not be advantageous during the training phase. This is because, at this stage, all modalities are available and inherently coherent, rendering the EBM somewhat superfluous for the SBM. However, during inference time,  in scenarios where multiple modalities are missing, the EBM becomes vital for the SBM. It aids in more effectively navigating the joint score landscape, thereby enhancing the model's performance in predicting more coherent output.
>
> *“Equation (5) doesn't make much sense”*
>
> Thank you for pointing out the error. We have updated the section.

---

> > ### Comment · Reviewer_qLtQ · 2023-11-23
> >
> > Thanks. I have read the rebuttal. Based on my understanding of this work, I would keep my rating

---

### Author Response · Authors · 2023-11-21
**Rebuttal by Authors**

Dear reviewers and chairs,

Thank you for your comments. We have made the following changes:

* We have added additional experiments on image and audio modalities from the MHD dataset by [1]. We selected two modalities from the dataset which are the image of MNIST images with audio recordings of each digit. The audio was transformed into a spectrogram format and paired with each digit of the 50,000 training set and 10,000 test set. We used the same architecture as [1] with a latent size of 64. We have put the results in Appendix Section A.8 comparing SBM-VAE and two of the baselines.

* We have added standard deviations to Table 1. Standard deviation was already present in the figures as a shade under the graphs (best viewed when zoomed)

* We have added Table 3, Figure 13, and Figure 14, which show the conditional performance of each modality and the average results

* Appendix section A.4.2. shows an ablation study on the effects of different beta values on the score model and performance of different beta baselines.

* Section A.5 discusses mode coverage experiments in unconditional generation which shows all modes have been covered in generations.

* We have also added experiments on finetuning the decoder which can be found in Appendix section A.5.1.

* The new changes are highlighted in red color in the paper so that they can be easily identified by reviewers.

[1] M. Vasco, H. Yin, F. S. Melo, and A. Paiva. Leveraging hierarchy in multimodal generative models for effective cross-modality inference. Neural Networks, 146:238–255, 2 2022.

---

### Meta-Review · Area_Chair_yybr · 2023-12-08

**Metareview:**

This paper presents a method for multi-modal representation learning consisting of two steps: the first steps learns initial representation for each view independently, while the second step couples initial representations from all views and learns a joint latent representation via score matching. The author presents good generation quality conditioned on a subset of views. There have been discussions on the experimental setup between authors and reviewers. My more critical concern is that the paper appears to have some conflicting intuitions, mainly due to the two-step approach. In the first step it is assumed that each latent variable z_k  only captures modality specific representations, and so depending on the latent variable dimension some information is lost. But then the second step is trying to learn a representation that captures correlation between the views (so we can hope to do better condition generation). I think it is possible that the first step may lose correlation that cannot be recovered in the second step. In fact in sec 2.2 the author described a coherence guidance term that aims to learn mutual information between the views leveraging the pairing info between the views. I suggest the authors to come up with a cleaner formulation with consistent intuitions.

**Justification For Why Not Higher Score:**

The intuitions behind the method are somewhat contradictory to each other.

**Justification For Why Not Lower Score:**

N/A

---

### Decision · Program_Chairs · 2024-01-16

Reject